# Chiral coordination polymer nanowires boost radiation-induced in situ tumor vaccination

Zhusheng Huang [1,2,3,6], Rong Gu[1,6], Shiqian Huang[2,6], Qian Chen[1], Jing Yan[4], Xiaoya Cui[5], Haojie Jiang[1], Dan Yao[1], Chuang Shen[2], Jiayue Su[5], Tao Liu[5], Jinhui Wu [1], Zhimin Luo [2,7] ✉, Yiqiao Hu [1,7] ✉ & Ahu Yuan [1,7] ✉

Radiation-induced in situ tumor vaccination alone is very weak and insufficient to elicit robust antitumor immune responses. In this work, we address this issue by developing chiral vidarabine monophosphate-gadolinium nanowires (aAGd-NWs) through coordination-driven self-assembly. We elucidate the mechanism of aAGd-NW assembly and characterize their distinct features, which include a negative surface charge, ultrafine topography, and right-handed chirality. Additionally, aAGd-NWs not only enhance X-ray deposition but also inhibit DNA repair, thereby enhancing radiation-induced in situ vaccination. Consequently, the in situ vaccination induced by aAGd-NWs sensitizes radiation enhances CD8[+] T-cell-dependent antitumor immunity and synergistically potentiates the efficacy immune checkpoint blockade therapies against both primary and metastatic tumors. The well-established aAGd-NWs exhibit exceptional therapeutic capacity and biocompatibility, offering a promising avenue for the development of radioimmunotherapy approaches.

In situ tumor vaccines involve the administration of antigens directly to the tumor site to stimulate antitumor immune responses[1–3]. During the last decade, notable research has highlighted the potential of the irradiated tumor itself as an intrinsic in situ vaccine, as evidenced by the occurrence of abscopal effects in clinical settings[4,5]. However, tumor cells have limited X-ray absorption capacity, and the tumor microenvironment lacks immune adjuvants necessary for effective phagocytosis and presentation of tumor antigens[6,7]. Consequently, radiation therapy (RT)-induced in situ vaccination has shown unsatisfactory results, with abscopal effects occurring in less than 1% of cases in clinical practice[8]. To overcome this challenge, high atomic number (high-Z) radiosensitization strategies involving the use of $HfO_2$ nanoparticles, AGuIX, and other approaches have been introduced to enhance X-ray deposition, offering modest improvements in RT-induced in situ vaccination[9–11]. Nevertheless, there remains a significant obstacle in achieving substantial augmentation of RT-induced in situ vaccination beyond the high-Z radiosensitization strategy to effectively enhance antitumor immunity. RT-induced in situ vaccination primarily involves the enhanced recognition and presentation of tumor antigens by the immune system[3,12]. The high-Z radiosensitization strategy aims to increase the generation of reactive oxygen species (ROS), leading to endoplasmic reticulum stress and subsequent induction of immunogenic cell death (ICD)[13–15]. This process involves the exposure of calreticulin (CRT) on the tumor cell surface, as well as the release of high mobility group box 1 (HMGB1) or adenosine triphosphate (ATP), which attract dendritic cells (DCs) for

[1]State Key Laboratory of Pharmaceutical Biotechnology, Medical School and School of Life Science, Nanjing University, Nanjing 210093, China. [2]State Key Laboratory for Organic Electronics and Information Displays (SKLOEID), School of Chemistry and Life Sciences, Nanjing University of Posts and Telecommunications, Nanjing 210023, China. [3]Cancer Centre and Institute of Translational Medicine, Faculty of Health Sciences, University of Macau, Macau SAR 999078, China. [4]The Comprehensive Cancer Centre of Drum Tower Hospital, Medical School of Nanjing University, Nanjing 210023, China. [5]Beijing Frontier Research Center for Biological Structures, School of Life Sciences, Tsinghua University, Beijing 100083, China. [6]These authors contributed equally: Zhusheng Huang, Rong Gu, Shiqian Huang. [7]These authors jointly supervised this work: Zhimin Luo, Yiqiao Hu, Ahu Yuan. ✉e-mail: iamzmluo@njupt.edu.cn; huyiqiao@nju.edu.cn; yuannju@nju.edu.cn

phagocytosis and antigen presentation[16]. Another critical aspect of ICD is the production of type I interferons (IFN-α and IFN-β), which act as danger-associated molecular patterns (DAMPs) and play a crucial role in linking innate and adaptive immune responses to initiate antitumor immunity[17]. Type I interferons have been shown to be essential for the cross-presentation of exogenous antigens by altering the acidification of antigen-presenting cell (APC) phagosomes, prolonging the retention time of antigens, and facilitating antigen release into the cytoplasm[18]. In vivo studies have demonstrated that mice lacking type I IFN receptors fail to reject highly immunogenic tumor cells[19,20]. However, our previous investigations have revealed that the High-Z strategy alone is insufficient in inducing the secretion of type I interferons, which may explain why high-Z strategies have limited efficacy in promoting RT-induced in situ vaccination[13,14].

Following radiation exposure, detectable fragments of DNA are released into the cytoplasm or extracellular spaces of tumor cells. These DNA fragments have the potential to be phagocytosed by surrounding immune cells, activating the cGAS/STING pathway and subsequently promoting the secretion of type I interferons[21–23]. However, the release of DNA after radiation is often insufficient, partially due to the presence of DNA repair mechanisms within tumor cells[24]. Upon radiation or high-Z radiation treatment, tumor cells promptly initiate multiple pathways to repair damaged DNA, aiming to maintain genomic integrity, which potentially leads to the inadequate secretion of type I interferons by surrounding immune cells[25]. Recent CRISPR-Cas9 screening studies have identified various DNA repair-related enzymes, such as nucleotide reductase, DNA polymerase, and DNA ligase, that contribute to radiation resistance[26–28]. Notably, these enzymes utilize dNTPs (e.g., dATP) or NTPs (e.g., ATP) as substrates or coenzymes[29]. Hence, a potential strategy involves employing competitive inhibitors to block the functions of dNTPs or NTPs, thereby simultaneously inhibiting these DNA repair-related enzymes. In this regard, vidarabine monophosphate (ara-AMP), a widely used clinical treatment for herpes simplex virus or herpes zoster virus infections, becomes relevant[30].

Following administration, ara-AMP undergoes sequential phosphorylation by kinases to form adenine arabinoside triphosphate (ara-ATP), an analog of dATP and ATP. Ara-ATP acts as a competitive inhibitor of the aforementioned DNA repair-related enzymes[31,32]. Therefore, we speculate that the combination of ara-AMP and the high-Z strategy could synergistically enhance radiation sensitivity, induce double-strand breaks, inhibit DNA repair, and promote the release of DNA fragments for cGAS/STING sensing.

In this work, we develop ara-AMP-Gd chiral nanowires (aAGd-NWs) through coordination-driven self-assembly based on vidarabine monophosphate (ara-AMP) and gadolinium (Gd), a clinical contrast agent. The extensively characterized aAGd-NWs exhibit a negative surface charge, ultrafine topography, and, notably, right-handed chirality. These features endow aAGd-NWs with the ability to penetrate deeply into tumor tissues, which forms the basis for their radiosensitization effect in damaging tumor cells located away from blood vessels. Additionally, we elucidate the mechanism of aAGd-NW assembly using cryo-electron microscopy (cryo-EM) in conjunction with 3D reconstruction. Specifically, Gd ions coordinate with the phosphate and primary amino groups of ara-AMP, facilitating the directional stacking process that results in the formation of ultrafine and chiral nanowires. As a result of the presence of the high-Z element Gd, aAGd-NWs exhibit efficient X-ray deposition, scattering, and emission capabilities, which enhance the generation of ROS. This process enhances the exposure of CRT on the surface of tumor cells and causes DNA double-strand breaks. Simultaneously, aAGd-NWs released ara-AMP, which acts as a DNA repair inhibitor, leading to the accumulation of DNA fragments within the cytoplasm, which promotes the activation of the cGAS/STING pathway and the subsequent secretion of interferon-beta (IFN-β) by surrounding immune cells. Consequently, the combined treatment of aAGd-NWs and radiation therapy synergistically induces in situ vaccination, facilitates immune priming, and potentiates the efficacy of checkpoint blockade immunotherapies (CBI) against both primary and metastatic tumors (Fig. 1).

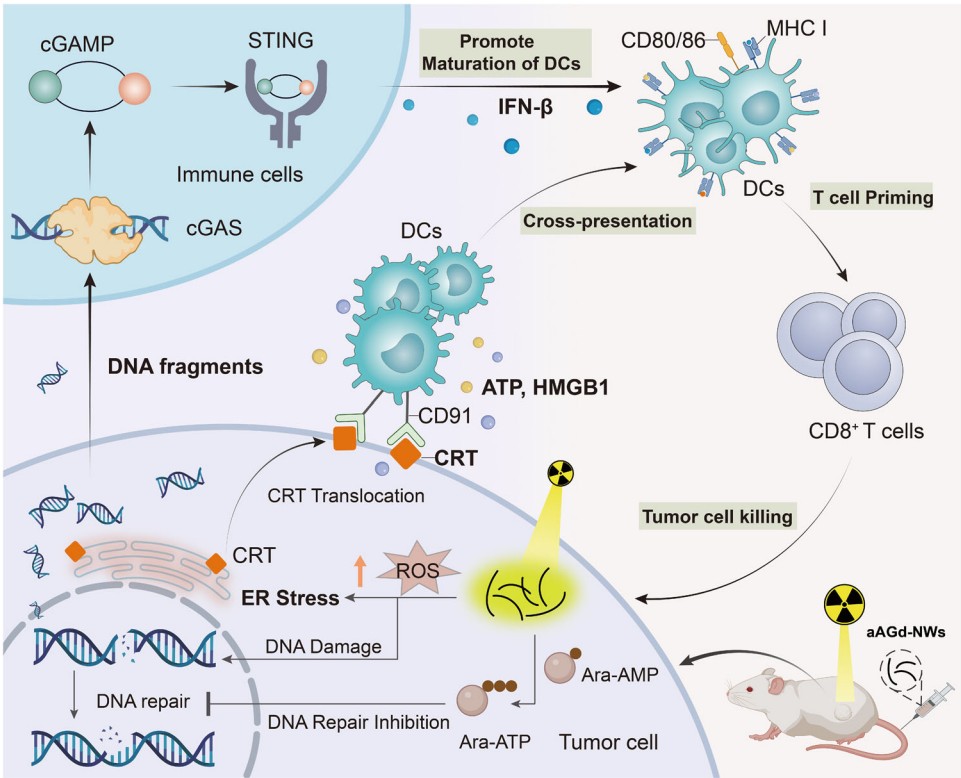

**Fig. 1 | Therapeutic mechanism of aAGd-NWs.** Schematic illustration of aAGd-NWs sensitized radiation for inducing potent in situ vaccination.

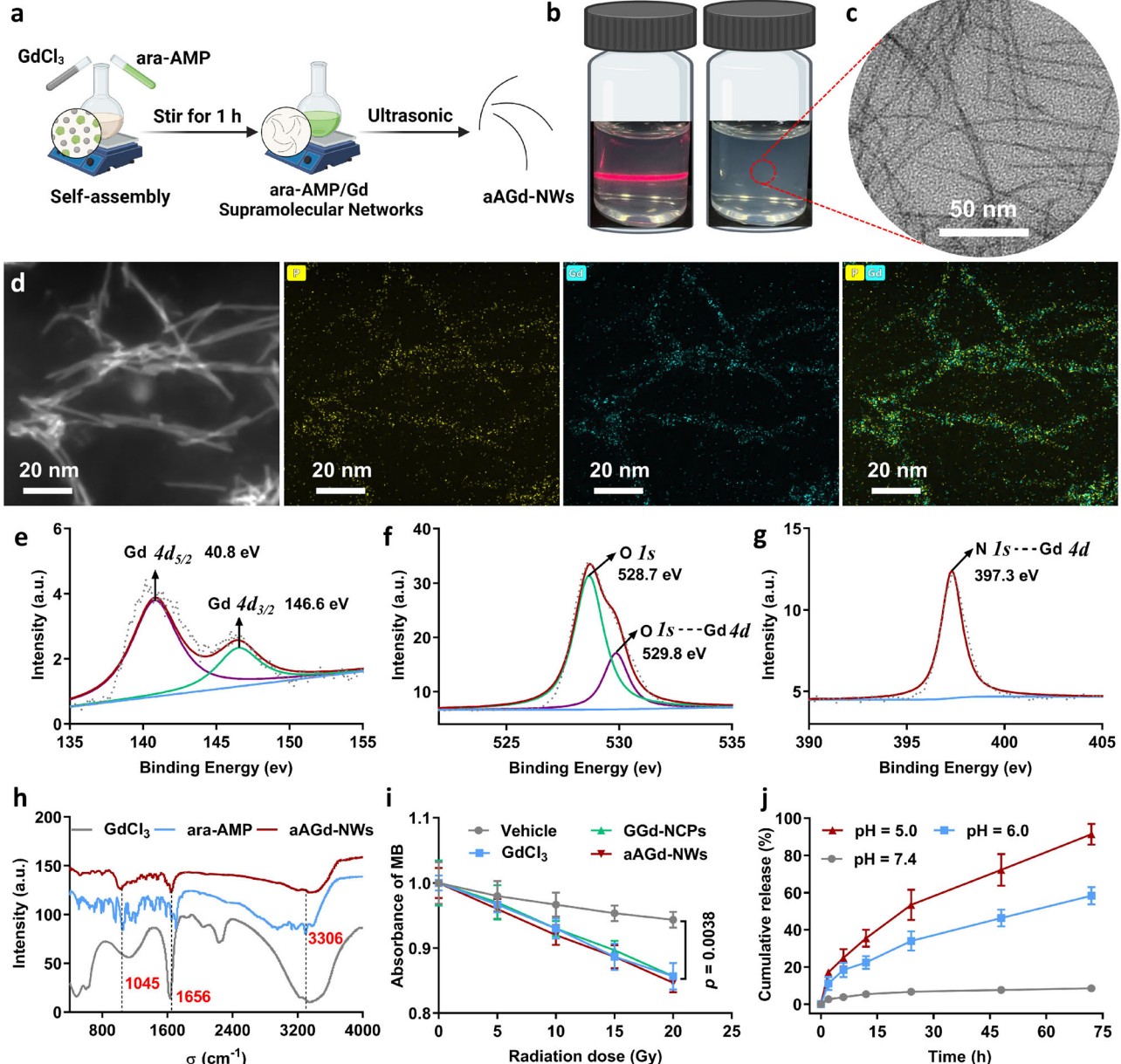

**Fig. 2 | Preparation and characterization of aAGd-NWs. a** Schematic depiction of the process for preparing aAGd-NWs. Created with BioRender.com. **b** The Tyndall effect and photographs of the obtained aAGd-NWs. **c** HRTEM imaging of aAGd-NWs, scale bar = 50 nm. This experiment was repeated twice independently with similar results. **d** EDS mapping of the aAGd-NWs, scale bar = 20 nm. This experiment was repeated twice independently with similar results. **e–g** High-resolution Gd *4d* (**e**), O *1s* (**f**), N *1s* (**g**) XPS spectra of aAGd-NWs, arbitrary units (a. u.). **h** FT-IR spectrum of GdCl$_3$, free ara-AMP and aAGd-NWs, arbitrary units (a. u.). (**i**) Assessment of hydroxyl radicals produced by Vehicle, GdCl$_3$, GGd-NCPs, and aAGd-NWs ([Gd] = 20 μM) under X-ray irradiation, as determined by the decay of MB ($n = 3$ experimental replicates). **j** Release profile of ara-AMP from aAGd-NWs in HEPES buffer containing 10% serum at different pH values (7.4, 6.0, 5.0) in vitro ($n = 3$ experimental replicates). All data were shown as mean ± SD. One-way ANOVA analysis of variance was used for multiple groups, and $p$ values < 0.05 indicate statistically significant. Source data are provided as a Source Data file.

## Results

### Preparation and characterization of aAGd-NWs

Gd and equimolar vidarabine monophosphate (ara-AMP) self-assembled to form white precipitates, which were subsequently washed and dispersed into opalescent NCPs via ultrasonication (Fig. 2a, b). Interestingly, transmission electron microscopy (TEM) imaging revealed that the resulting coordination polymers exhibited an ultrafine nanowire morphology, referred to as ara-AMP/Gd nanowires (aAGd-NWs), with a width of approximately 1.3 nm and lengths ranging from 50 to 200 nm (Fig. 2c). High-resolution TEM (HRTEM) and electron diffraction analyses demonstrated the amorphous nature

of these ultrafine nanowires, as evidenced by the absence of crystalline fringes (Supplementary Fig. 1a, b). Furthermore, powder X-ray diffraction (PXRD) analysis also confirmed the amorphous state of the aAGd-NWs (Supplementary Fig. 1c). High-resolution TEM energy dispersive spectroscopy mapping (EDS mapping) confirmed the presence of Gd and P within the aAGd-NWs (Fig. 2d), and the high-resolution Gd *4d* X-ray photoelectron spectroscopy (XPS) spectrum exhibited binding energies at 140.8 eV and 146.6 eV, corresponding to Gd *4d$_{5/2}$* and Gd *4d$_{3/2}$*, respectively (Fig. 2e). The high-resolution O *1s* and N *1s* XPS spectra revealed shifts in binding energies (from 531.1 eV to 528.7 eV for O *1s* and from 397.9 eV to 397.3 eV for N *1s*), indicating

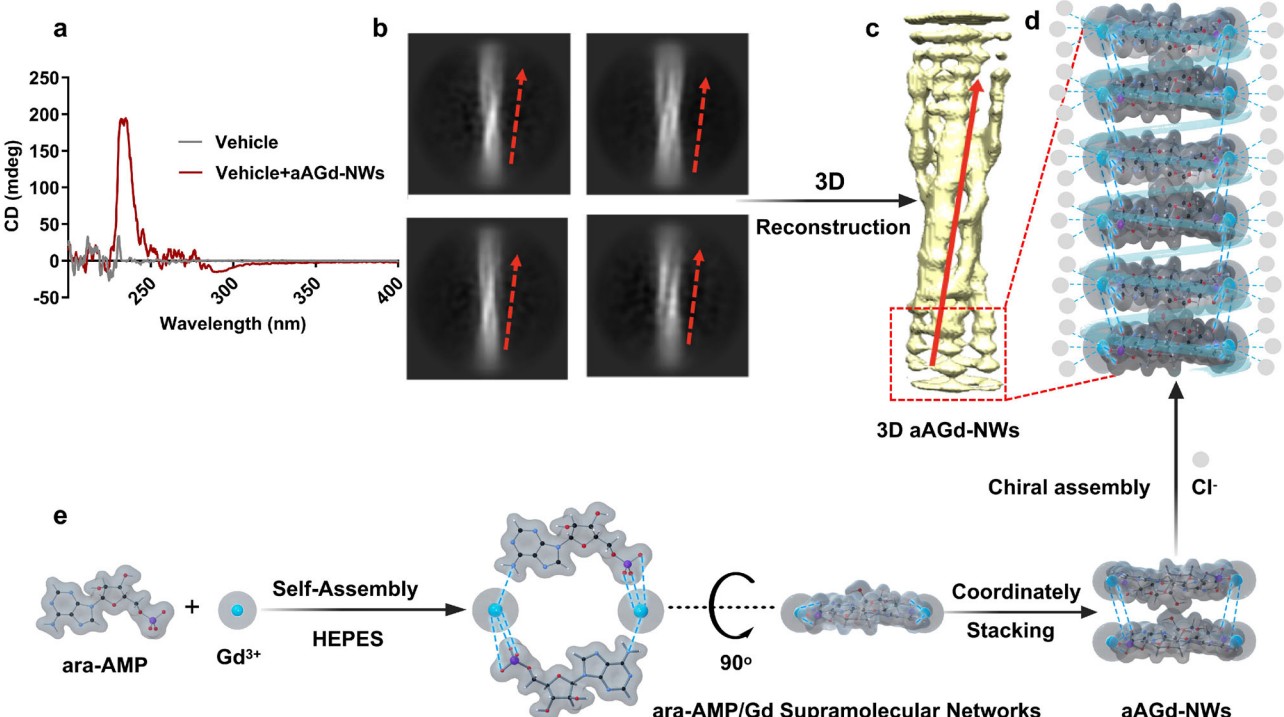

**Fig. 3 | Structural analysis of aAGd-NWs. a** Circular dichroism (CD) spectrum of aAGd-NWs. **b** Calculated images of aAGd-NWs based on 2D imaging. This experiment was repeated twice independently with similar results. **c** Calculated 3D structure based on 3D modeling reconstruction. **d, e** Proposed self-assembly mechanism of aAGd-NWs. Source data are provided as a Source Data file.

coordination between Gd-O and Gd-N elements, respectively (Fig. 2f, g and Supplementary Fig. 1d, e). The coordination effects between ara-AMP and Gd were further demonstrated by Fourier transform infrared spectroscopy (FT-IR), and aAGd-NWs presented characteristic absorption peaks of GdCl₃ (~1656 cm⁻¹) and ara-AMP (~1045 and ~3306 cm⁻¹), respectively (Fig. 1h). Statistical analysis based on HRTEM data indicated that aAGd-NWs exhibited an average width of 1.3 ± 0.2 nm (Supplementary Fig. 1f), while zeta (ζ) potential measurements revealed a negatively charged surface with a ζ-potential of −12.43 mV for aAGd-NWs (Supplementary Fig. 1g). The average hydrated particle size (Dynamic Light Scattering, DLS) of aAGd-NWs is 189 nm (Supplementary Fig. 1h), aligning with the length observed through HRTEM, and exhibit relatively high stability in water (25 °C), PBS (25 °C) and 10% FBS (37 °C) (Supplementary Fig. 1i). Moreover, the UV absorption spectrum remains consistent after 72 h compared to the initial spectrum, indicating their stability, as evidenced in Supplementary Fig. 1j.

For comparison, spherical particles of GGd-NCPs (5′-GMP-Gd nanoscale coordination polymers, prepared in our previous study[13]) were synthesized by coordinating Gd with 5′-guanosine monophosphate (5′-GMP) as a nonfunctional ligand. We then evaluated the production of hydroxyl radicals (•OH) by vehicle, GdCl₃, GGd-NCPs and aAGd-NWs at different radiation doses of 0, 5, 10, 15, and 20 Gy. The results indicated that the introduction of the high atomic number element gadolinium (GdCl₃, GGd-NCPs and aAGd-NWs) effectively enhanced X-ray deposition and significantly promoted the generation of •OH, leading to the accelerated decay of methylthionine (MB) (Fig. 2i)[33]. Subsequently, we further investigated the formation of other ROS using electron spin resonance (ESR) spectroscopy. TEMP was used as a trap for singlet oxygen (¹O₂) to produce TEMPO, which was quantified by ESR spectroscopy. RT can induce the production of a measurable amount of singlet oxygen, yet aAGd-NWs exert no significant influence on this process (Supplementary Fig. 1k). The results obtained are in alignment with previously reported findings[34].

Moreover, the release profile of ara-AMP from aAGd-NWs was assessed in HEPES buffer supplemented with 10% serum at varying pH values (Fig. 2j). The release rate of ara-AMP from aAGd-NWs was found to be minimal at pH 7.4 but notably accelerated at pH 6.0 and 5.0, suggesting that aAGd-NWs could responsively release ara-AMP upon their penetration into tumor tissues or cellular lysosomes.

## Structural analysis of aAGd-NWs

Subsequently, the stoichiometric ratio of Gd and ara-AMP in the self-assembly process was determined to be approximately 1:1 through XPS elemental analysis of freeze-dried powder of nondispersed ara-AMP-Gd coordination polymers (Supplementary Fig. 1l). Circular dichroism (CD) spectroscopy, a method based on the differential absorption of left and right circularly polarized light, was employed to investigate the chirality of aAGd-NWs. Strikingly, the CD spectrum revealed that the aAGd-NWs had a right-handed chiral structure, exhibiting a mean residue ellipticity of 190 mdeg (Fig. 3a). The presence of chirality in aAGd-NWs indicated ordered growth and prompted further investigation of their structure and assembly mechanism using cryo-electron microscopy (cryo-EM). Cryo-EM 2D images of aAGd-NWs from different views were collected for 3D modeling reconstruction (Supplementary Fig. 2a–e and Supplementary Table 1). The fitting diagram of the 2D images, as well as the calculated 3D models based on electron cloud density, unequivocally confirmed the right-handed chirality of aAGd-NWs, in complete agreement with the CD spectroscopy results (Fig. 3b, c and Supplementary Movie 1). Based on these simulation results, we propose a plausible assembly mechanism for aAGd-NWs: ara-AMP and Gd initially self-assemble into supramolecular macrocyclic structures, which subsequently stack in a sequential manner via coordination-driven self-assembly. During the stacking process, the assembling macrocyclic structure undergoes a 180° rotation and coordinates with the existing structures, resulting in the observed right-handed chirality of aAGd-NWs. Then, the addition of 2% MSA (dissolved in 0.9% NaCl), serving as emulsifier and aggregation

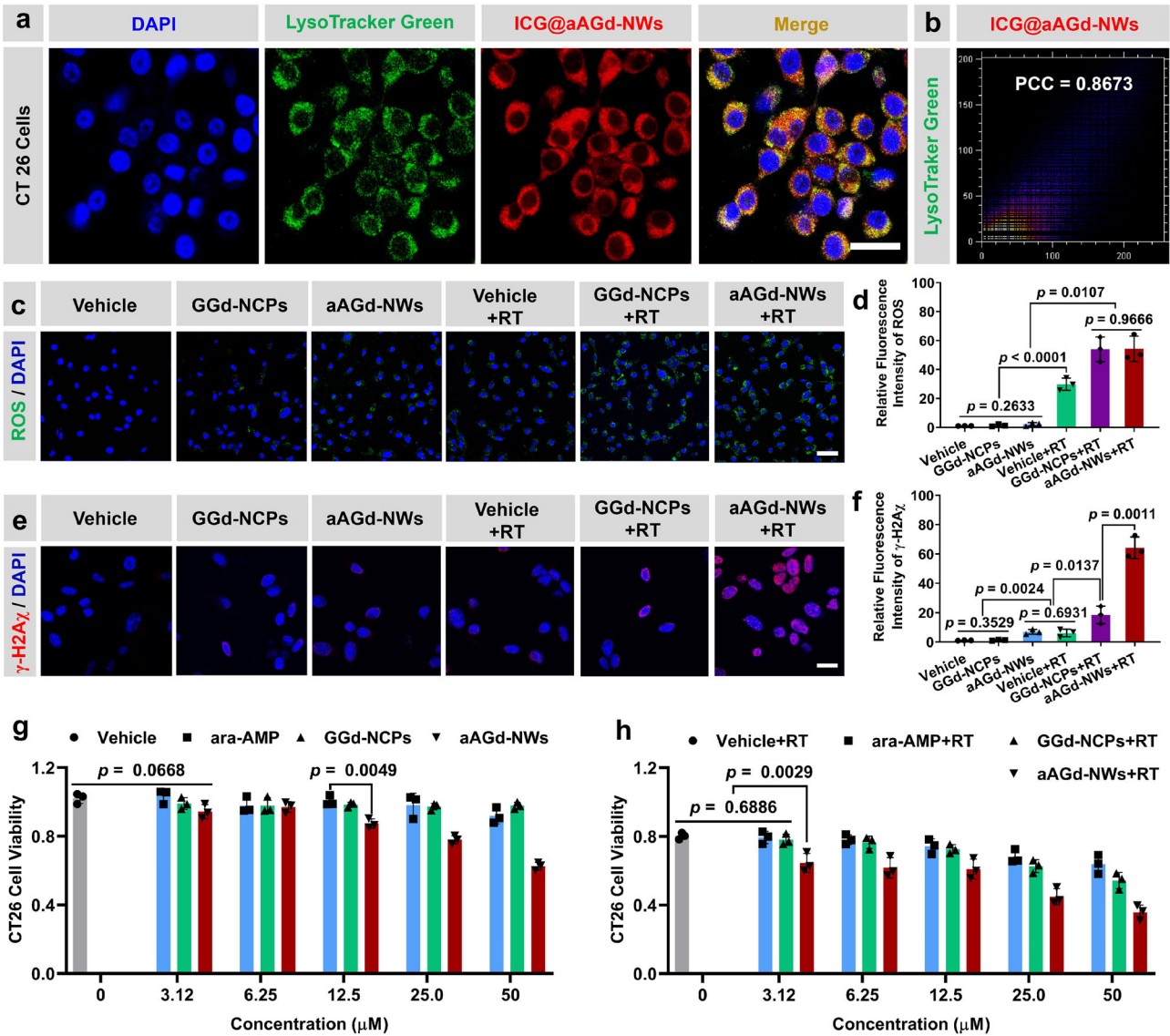

**Fig. 4 | Radiosensitization of aAGd-NWs in vitro. a** Confocal laser scanning microscope (CLSM) images of CT26 cells stained with DAPI, LysoTracker Green and ICG@aAGd-NWs, respectively. The co-localization of aAGd-NWs within lysosomes is indicated by the yellow regions, scale bar = 20 µm. This experiment was repeated twice independently with similar results. **b** Pearson correlation colocalization (PCC) analysis of treated CT26 tumor cells by ImageJ (Colocalization Finder).
**c** Intracellular generation of ROS in CT26 cells detected by ROS probe $H_2DCFDA$ (green fluorescence) merged with DAPI (blue fluorescence), scale bar = 50 µm. This experiment was repeated twice independently with similar results. **d** Quantification of relative fluorescent signal based on (**c**) by ImageJ ($n = 3$ experimental repeats).

**e** Evaluation of intracellular γ-H2Aχ in CT26 tumor cells, scale bar = 10 µm. This experiment was repeated twice independently with similar results. **f** Quantification of relative fluorescence based on (**e**) by ImageJ ($n = 3$ experimental repeats).
**g, h** Cytotoxicity assessment of Vehicle, GGd-NCPs and aAGd-NWs ([Gd] = 0, 3.12, 6.25, 12.5, 25.0, 50.0 µM, $n = 3$ experimental repeats) against CT26 cells without (**f**) and with (**g**) X-ray irradiation (5 Gy × 1). All data were shown as mean ± SD. Statistical analysis was performed using two-tailed Student's $t$-test for comparing two groups, and one-way ANOVA analysis of variance for multiple groups. $p$ values > 0.05 represented nonsignificance (N.S.), while $p$ values < 0.05 indicate statistical significant. Source data are provided as a Source Data file.

inhibitor, lead to uniformly dispersed and stable aAGd-NWs. Consequently, it is reasonable to postulate that $Cl^-$ is also involved in the coordination of Gd (Fig. 3d, e).

**In vitro radiosensitization of aAGd-NWs**
To assess the intracellular distribution of aAGd-NWs, we incorporated indocyanine green (ICG) into aAGd-NWs during the coordination process, yielding ICG@aAGd-NWs. Subsequently, CT26 tumor cells treated with ICG@aAGd-NWs (displaying red fluorescence), along with DAPI (blue fluorescence) and LysoTracker (green fluorescence), were examined using confocal laser scanning microscopy (CLSM). The imaging results unequivocally demonstrated the effective cellular uptake of ICG@aAGd-NWs by CT26 tumor cells, with predominant

localization within lysosomes (Fig. 4a). Further analyses quantified the degree of colocalization, as illustrated in Fig. 4b, resulting in a Pearson colocalization coefficient (PCC) of 0.8673. To investigate the in vitro radiosensitizing effect of aAGd-NWs, we utilized 2,7-dichlorodihydro-fluorescein diacetate ($H_2DCFDA$) to monitor intracellular ROS generation. The green fluorescence emitted by $H_2DCFDA$ within CT26 tumor cells subjected to radiation sensitization with aAGd-NWs or GGd-NCPs was markedly stronger than that in cells treated with radiation alone. This observation indicated that the high-Z strategy involving aAGd-NWs effectively facilitated X-ray deposition, leading to enhanced intracellular ROS production within tumor cells (Fig. 4c, d and Supplementary Fig. 3a). Furthermore, we employed anti-γ-H2Aχ staining to quantitatively assess DNA double-strand breaks. Notably,

aAGd-NW-sensitized radiation therapy induced a higher number of DNA DSBs within the nucleus of tumor cells, surpassing the levels observed with radiation alone or the high-Z radiosensitization achieved by GGd-NCPs (Fig. 4e, f and Supplementary Fig. 3b). These outcomes can be attributed to the significant synergistic effects of aAGd-NWs, which amplified DNA damage and hindered DNA repair processes.

Subsequently, we conducted an assessment of the therapeutic cytotoxicity of aAGd-NWs, with or without radiation, against CT26 tumor cells. Initially, we observed that GGd-NCPs did not induce significant cytotoxicity in CT26 tumor cells, likely due to the presence of the nonfunctional ligand 5′-GMP. Conversely, aAGd-NWs exhibited pronounced inhibition of CT26 tumor cell growth, which can be attributed to the inhibition of DNA synthesis/repair caused by the released ara-AMP (Fig. 4g). Furthermore, when combined with irradiation, aAGd-NWs demonstrated superior inhibitory capacity compared to free ara-AMP or GGd-NCPs (Fig. 4h), indicating the synergistic therapeutic properties of aAGd-NWs against tumor cells. The flow cytometry results of PI and Annexin V staining indicate that aAGd-NW-sensitized RT promotes more apoptosis, suggesting that the aAGd-NWs significantly enhance the efficacy of RT in killing tumor cells (Supplementary Fig. 4).

## aAGd-NWs deeply penetrated 3D tumor spheroids

To evaluate the penetrating capability of ultrafine aAGd-NWs in comparison to spherical nanoparticles, we investigated their penetration profiles in 3D CT26 spheroids[35–37]. For this purpose, indocyanine green (ICG) was incorporated as a fluorescent probe into spherical GGd-NCPs (nanoparticles) and aAGd-NWs (nanowires). Following a 6-hour coincubation, CT26 3D spheroids treated with ICG@GGd-NCPs exhibited weak red fluorescence, confined mainly to the marginal regions, suggesting limited tissue penetration. In contrast, the ultrafine ICG@aAGd-NWs displayed stronger red fluorescence, penetrating deeply into 3D spheroids (>50 μm, Fig. 5a). The dynamic fluorescence intensity at the 30 μm section further confirmed the superior penetration capacity of ultrafine aAGd-NWs compared to spherical GGd-NCPs (Fig. 5b). Due to the light absorption and scattering properties inherent in most biological samples, the penetration depth of confocal laser microscopy is typically limited to approximately 100 μm[38]. To delve deeper into the penetration capabilities of ultrafine aAGd-NWs in CT26 3D spheroids, we dissociated these spheroids into single cells for flow cytometry analysis. As illustrated in Supplementary Fig. 5, compared to vehicle-treated 3D cell spheroids, the uptake rate of ICG@aAGd-NWs in tumor cells within 3D spheroids was significantly higher at 79.7%. These findings suggest that ultrafine aAGd-NWs possess the capacity to deeply penetrate the spheroids and effectively infiltrate a substantial proportion of tumor cells. Subsequently, we examined the therapeutic cytotoxicity of aAGd-NW-sensitized radiation therapy (RT) in CT26 3D spheroids using live/dead (calcein AM/PI) staining. Due to their excellent penetration, X-ray deposition and DNA repair inhibition, aAGd-NW-sensitized RT led to a significantly higher number of PI$^+$ dead tumor cells within 3D spheroids than GGd-NCP-sensitized RT or RT alone (Fig. 5c, d). Furthermore, aAGd-NWs demonstrated a good radiosensitization effect, maintaining optimal inhibition of tumor growth for ten consecutive days (Fig. 5e–g).

## Pharmacokinetics and radiosensitization efficacy evaluation in vivo

To detect the biodistribution of aAGd-NWs in vivo, we conducted pharmacokinetic analyses of plasma and tumor tissues (Fig. 6a). Initially, we compared the in vivo pharmacokinetics of free ara-AMP and aAGd-NWs at various time points (0.5, 1, 3, 6, 12, 24, 48 h). The blood elimination half-life (t$_{1/2}$) of free ara-AMP was approximately

3.52 ± 0.46 h, while aAGd-NWs exhibited a significantly prolonged half-life of approximately 7.28 ± 0.55 h (Fig. 6b and Supplementary Table 2). Subsequently, after injection of free ara-AMP or aAGd-NWs ([ara-AMP] = 34.7 mg kg$^{-1}$), we collected tumor tissues at 2, 6, 12, 24, 48, and 72 h to evaluate ara-AMP and aAGd-NWs accumulation. As depicted in Fig. 6c, aAGd-NWs-treated tumors exhibited the highest concentration of accumulated ara-AMP (53.9 μg g$^{-1}$ tumor tissue) at 6 h post administration. However, the accumulation time and concentration of free ara-AMP within tumor tissues were relatively short and low, with a concentration of 23.6 μg g$^{-1}$ observed at 2 h post administration. Furthermore, we quantified the concentration of Gd within tumor tissues at different time points by inductively coupled plasma optical emission spectrometer (ICP-OES). Upon incineration and nitrification of the tumor tissues, Gd accumulation was observed within the tumor tissues in the aAGd-NWs group, reaching the highest concentration of 23.0 μg g$^{-1}$ tumor tissues at 6 h, followed by gradual elimination over the subsequent 48 h (Fig. 6d). These results clearly demonstrated that compared to free ara-AMP, aAGd-NWs exhibit a prolonged half-life and superior tumor accumulation capacity, suggesting their essential role in enhancing radiosensitization. We then performed a detailed analysis of aAGd-NWs distribution in major organs using ICP-OES. As depicted in the Supplementary Fig. 6, 6 h post aAGd-NWs injection, the highest concentration was observed in tumor tissues, followed by distribution in renal tissues, with smaller quantities detected in the heart, liver, spleen, and lungs. Importantly, the presence of free Gd ions could not be directly detected within tissues that have not been incinerated and nitrified. This finding indicates that post metabolic processes in vivo, Gd remains either coordinated or partially coordinated state rather than in a free state, which implies a preliminary indication of biological safety.

Subsequently, we evaluated the radiosensitization efficacy of aAGd-NWs in CT26 tumor-bearing mice. Once the tumor volume reached 80–100 mm³, all mice were randomly assigned to 8 groups, including Vehicle, ara-AMP, GGd-NCPs, and aAGd-NWs with or without RT (5 Gy). The treatments and irradiation were performed on Day 0 and Day 6, respectively. As depicted in Supplementary Fig. 7a–d, the presence of the high-Z element Gd within GGd-NCPs did not significantly inhibit tumor growth. Treatment with free ara-AMP also did not exert a noticeable tumor inhibitory effect, which can likely be attributed to its limited tumor accumulation capacity. However, following the administration of aAGd-NWs, which exhibited improved tumor accumulation capacity, a moderate tumor-suppressive effect was observed. Even when combined with radiation therapy (RT), free ara-AMP did not demonstrate any synergistic therapeutic effects, potentially explaining its lack of clinical use for radiosensitization. In comparison, GGd-NCPs, which sensitized radiation through the presence of the high-Z element, showed noticeable tumor inhibitory effects when combined with RT. Surprisingly, when high-Z radiosensitization was combined with DNA repair inhibition, aAGd-NWs exhibited the most significant radiosensitization capacities and long-lasting tumor suppression of tumor growth (Fig. 6e and Supplementary Fig. 7e, f). Tumor growth inhibition in the CT26 colorectal model was further confirmed by the weights of the excised tumor tissues on Day 14 for the vehicle group without irradiation or Day 18 for the other groups with irradiation (Fig. 6f). Immunofluorescence staining of γ-H2Aχ revealed that aAGd-NW-sensitized RT induced the highest number of fluorescent foci, indicating extensive DNA double-strand breaks in vivo (Supplementary Fig. 8 and Fig. 6g). TUNEL fluorescence staining and Ki67 immunohistochemical (IHC) staining also demonstrated the highest levels of apoptotic tumor cells and the lowest levels of proliferative tumor cells in the group of aAGd-NW-sensitized RT (Supplementary Fig. 8 and Fig. 6h, i). These results confirmed that aAGd-NWs could synergistically enhance radiation sensitivity through increased X-ray deposition and subsequent inhibition of DNA repair.

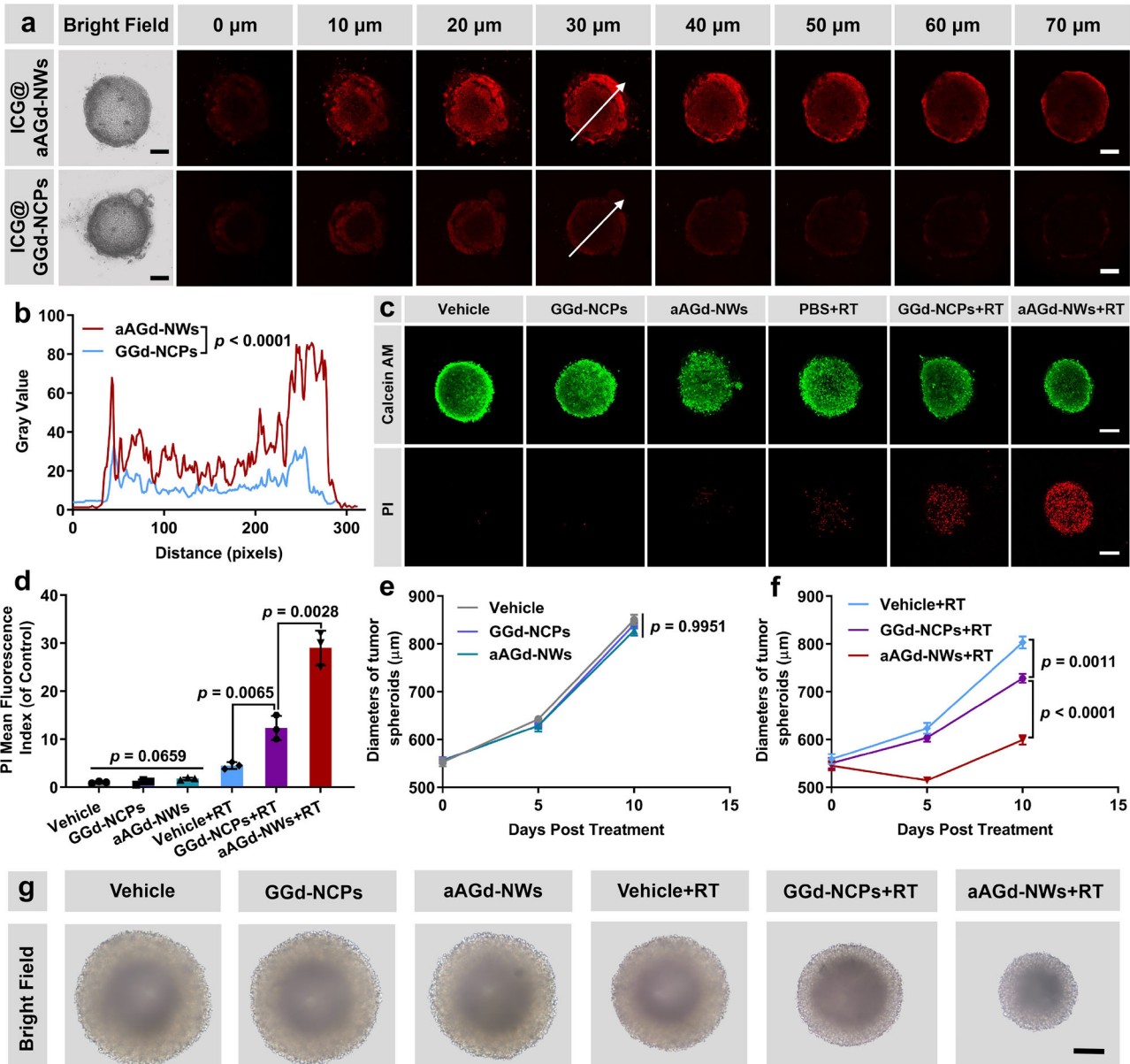

**Fig. 5 | Penetration and therapeutic efficacy of aAGd-NW-sensitized RT on CT26 3D spheroids.** **a** Penetration profiles of ICG@GGd-NCPs and ICG@aAGd-NWs within CT26 3D spheroids, scale bar = 200 μm. This experiment was repeated twice independently with similar results. **b** Dynamic curves illustrating the fluorescence intensity of ICG along the white arrow. **c** Calcein AM/PI staining of CT26 3D spheroids treated by Vehicle, GGd-NCPs and aAGd-NWs with or without RT (5 Gy), scale bar = 200 μm. Live cells are labeled green with Calcein AM, and dead cells are labeled red by PI. This experiment was repeated twice independently with similar results. **d** Relative mean fluorescence index of PI based on (**c**) (*n* = 3 experimental repeats). **e**, **f** Growth curves of CT26 3D spheroids following treatment with Vehicle, GGd-NCPs and aAGd-NWs without (**e**) or with (**f**) RT (5 Gy) (*n* = 3 experimental repeats). **g** Representative images of CT26 3D spheroids on day 10 after various treatments, scale bar = 400 μm. This experiment was repeated twice independently with similar results. All data are shown as mean ± SD. Statistical significance was determined using two-tailed Student's *t*-test for pairwise comparisons, and one-way ANOVA analysis of variance for multiple groups. *p* values > 0.05 were considered non-significant (N.S.), while *p* values < 0.05 were considered statistically significant. Source data are provided as a Source Data file.

## Enhanced ICD induction for in situ vaccination

Next, we assessed the DNA-driven immune response and induction of ICD triggered by aAGd-NW-sensitized radiation. To determine whether damaged DNA fragments were released into the cytoplasm, we performed a PicoGreen assay, which uses a fluorescent dye that binds to double-stranded DNA, on CT26 tumor cells[39,40]. As depicted in Fig. 7a, b, within the first 6 h postradiation therapy (RT), minimally damaged DNA was detected in the cytoplasm across all groups. However, at 24 h post RT, numerous micronuclei, consisting of accumulated damaged DNA, were observed outside the nucleus in the aAGd-NWs+RT group (Fig. 7a, c). These findings indicated that aAGd-NW-sensitized radiation could enhance mitotic catastrophe and

facilitate the release of damaged DNA into the cytoplasm or extracellular spaces. In immune cells, cyclized GMP-AMP synthase (cGAS) recognizes DNA fragments and catalyzes the production of 2′,3′-cGAMP, which activates the downstream stimulator of interferon genes (STING) pathway, leading to the secretion of type I interferons[41-43]. To examine whether these released tumor-derived DNA fragments could be internalized by surrounding immune cells to activate their cGAS-STING pathway and induce type I interferon secretion, we conducted coculture experiments using aAGd-NWs+RT-treated CT26 tumor cells and RAW264.7 cells. As illustrated in Fig. 7d, aAGd-NWs+RT-treated CT26 tumor cells significantly up-regulated the phosphorylation levels of STING and interferon regulatory Factor 3

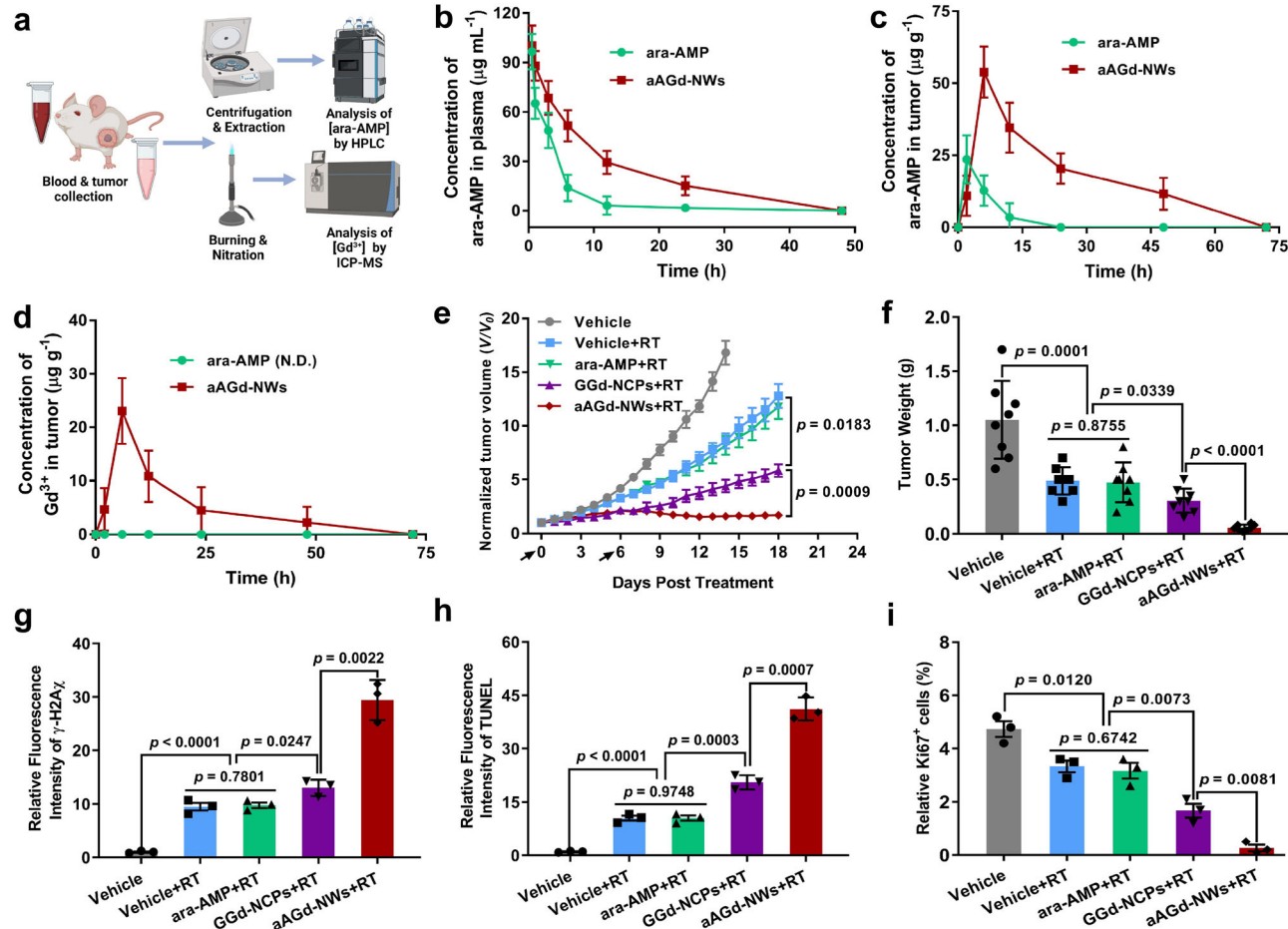

**Fig. 6 | Pharmacokinetics and radiosensitization efficacy of aAGd-NWs in vivo.**
**a** Schematic illustration of pharmacokinetic profiles. Created with BioRender.com.
**b** Pharmacokinetic profiles of free ara-AMP and aAGd-NWs in vivo ($n = 3$ mice).
**c**, **d** The dynamic concentrations of ara-AMP and Gd accumulated within tumor tissues determined by HPLC (**c**) and ICP-OES (**d**), respectively ($n = 3$ mice). Non-detectable (N.D.). **e** Normalized tumor growth curves of Vehicle, free ara-AMP, GGd-NCPs, and aAGd-NWs treatments with X-ray irradiation ($5\,Gy \times 2$ with fractions delivered 6 days apart), respectively ($n = 8$ mice). **f** Tumor weights after Vehicle, ara-

AMP, GGd-NCPs, and aAGd-NWs treatments with irradiation, respectively ($n = 8$ mice). **g**–**i** Quantification of γ-H2Aχ (**g**), TUNEL (**h**) mean fluorescent intensity, and relative percentage of Ki67 positive cells (**i**) after different treatments ($n = 3$ mice). All data were shown as mean ± SD. Statistical significance was determined using two-tailed Student's *t*-test for pairwise comparisons, and one-way ANOVA analysis of variance for multiple groups. *p* values > 0.05 were considered non-significant (N.S.), while *p* values < 0.05 were considered statistically significant. Source data are provided as a Source Data file.

(IRF-3) in the co-cultured cells. However, co-culture of ara-AMP or RT treated CT26 tumor cells with immune cells led to relatively weak STING and IRF-3 phosphorylation. This observation indicates that ara-AMP itself possesses a relatively weak capability to activate the cGAS/STING pathway, comparable to radiation therapy (Supplementary Fig. 9). Moreover, aAGd-NWs+RT markedly enhanced the secretion of INF-β by RAW264.7 cells (Fig. 7e), indicating the strong synergistic effects between aAGd-NWs and radiation.

In addition to IFN-β, we further evaluated other markers of ICD, such as CRT and ATP, induced by radiation therapy sensitized with aAGd-NWs[16,17]. CRT, which translocates to the cellular membrane surface, acts as an "eat me" signal for phagocytosis by APCs[44]. As depicted in Fig. 7f, exposure of CRT on the cell surface was very limited when RT was administered alone. However, when sensitized by GGd-NCPs, evident CRT translocation to the surface of treated tumor cells was observed. Moreover, aAGd-NWs sensitized radiation significantly induced more CRT exposure than GGd-NCPs sensitized radiation or radiation alone (Fig. 7g). In addition to DNA fragment release, the extensive DNA damage caused by aAGd-NW-sensitized radiation also activates the endoplasmic reticulum (ER) stress-dependent unfolded protein reaction (UPR), leading to enhanced CRT exposure[21]. Importantly, treatment with aAGd-NWs+RT also

significantly promoted HMGB1 release (Fig. 7h) and ATP secretion (Fig. 7i) from the treated CT26 tumor cells, which can recruit and activate APCs. As illustrated in Supplementary Fig. 10a–d, the capability of doxorubicin and oxaliplatin (classic ICD inducers) to induce ICD was comparable to RT alone, yet significantly inferior to aAGd-NW-sensitized RT. These findings indicated that aAGd-NWs can effectively sensitize radiation to enhance the immunogenicity of treated tumor cells, thereby favorably facilitating subsequent in situ vaccination induction.

## Systemic anti-tumor immune responses

Considering the good ability of aAGd-NWs+RT-treated tumor cells to induce ICD, we hypothesized that these cells, with increased immunogenicity, would further enhance systemic antitumor immune responses. To validate this hypothesis, we conducted prophylactic vaccination experiments in vivo using CT26 tumor cells. After treatment with RT or aAGd-NWs+RT, CT26 cell fragments were obtained by cryo-shocking in liquid nitrogen to eliminate their tumorigenic capacity while preserving their major structures. Subsequently, these tumor cell fragments or PBS were subcutaneously inoculated into the left side of healthy mice, and the immunization process was repeated 3 times following the procedures outlined in Fig. 8a. On Day 9, live CT26

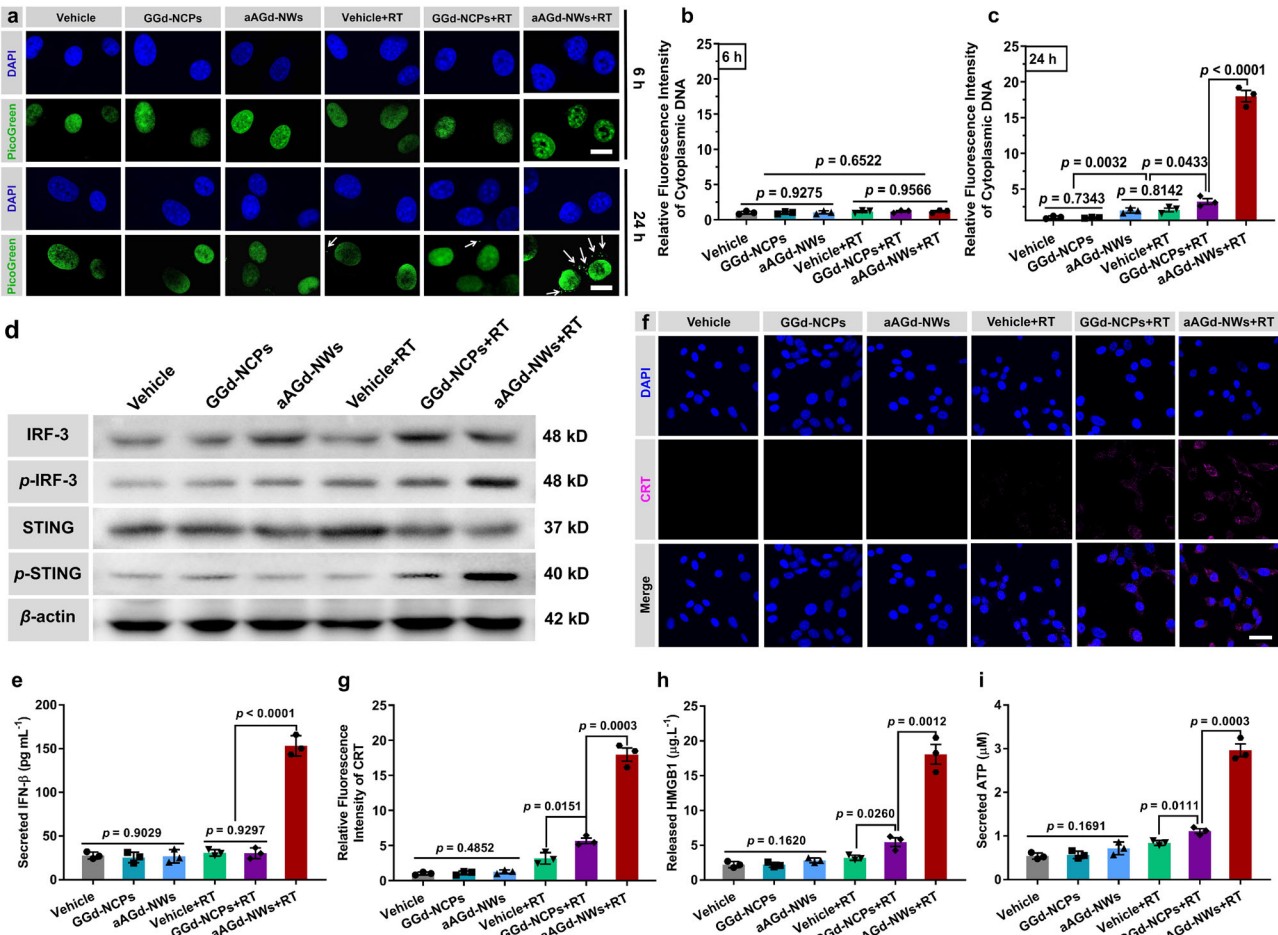

**Fig. 7 | Enhanced DNA damage and ICD induction. a** Representative images of DAPI and PicoGreen co-stained CT26 cell nuclei after various treatments, scale bar = 20 μm. The white arrows indicated damaged DNA fragments. This experiment was repeated twice independently with similar results. **b, c** Quantification of relative fluorescent intensity of cytoplasmic DNA by PicoGreen staining at 6 h (**b**) and 24 h (**c**) post X-ray irradiation, respectively (n = 3 experimental repeats). **d** Western blot of IRF3, p-IRF3, STING, p-STING, and β-actin as an internal reference. This experiment was repeated twice independently with similar results. **e** Detection of secreted IFN-β by ELISA kit (n = 3 experimental repeats). **f** Immunofluorescence of CT26 cells stained with anti-CRT antibody, scale bar = 20 μm. This experiment was repeated twice independently with similar results. **g** Quantification of relative CRT mean fluorescent intensity (n = 3 experimental repeats). **h, i** Detection of cytoplasmic HMGB1 (**h**) and ATP secretion (**i**) by ELISA kit and luciferin-based ATP assay kit (n = 3 experimental repeats). All data were shown as mean ± SD. Two-tailed Student's t-test was used to calculate statistical differences between two groups, and one-way ANOVA analysis of variance was used for multiple groups. p values > 0.05 represented nonsignificance (N.S.) and p values < 0.05 represented statistically significant. Source data are provided as a Source Data file.

cells were inoculated on the contralateral side of the mice, and tumor growth was monitored over the following period. As illustrated in Fig. 8b–d and Supplementary Fig. 11a, b, the prophylactic vaccination induced by aAGd-NWs+RT significantly inhibited CT26 tumor growth, resulting in 87.5% (7/8) tumor-free survival, which was markedly different from the two control groups.

We further established B16-OVA subcutaneous tumor model to investigate the effect of aAGd-NWs+RT on the presentation of the $OVA_{257-264}$ antigen (SIINFEKL). We observed that the proportion of $CD11c^+$ H2Kb-$SIINFEKL^+$ DCs from tumor draining lymph nodes (TDLNs) following the treatment of aAGd-NWs+RT was significantly higher compared to the Vehicle, GGd-NCPs groups, with or without RT (Fig. 8e and Supplementary Fig. 12). Additionally, to delve deeper into the effects of ICD, we analyzed the maturation of DCs in the TDLNs using flow cytometry. As depicted in Fig. 8f, a significantly higher proportion of mature DCs ($CD80^+$ $CD86^+$ gated within $CD11c^+$) appeared in TDLNs treated with aAGd-NWs+RT than in those treated with GGd-NCPs+RT or RT alone (Supplementary Fig. 13). This finding confirmed that aAGd-NWs sensitized RT synergistically promoted in situ vaccination, potentially enhancing systemic antitumor immune responses.

Furthermore, we evaluated the systemic antitumor therapeutic effects of aAGd-NW-sensitized radiation in a CT26 colorectal tumor model. As illustrated in Fig. 8g, and Supplementary Figs. 14−16, compared to RT alone, aAGd-NW-sensitized radiation effectively suppressed the growth of CT26 tumors. Considering the induction of in situ vaccination, we investigated whether the combined therapeutic effect of aAGd-NWs+RT was associated with the activation of T-cell-dependent immune responses. We examined the infiltration of $CD4^+$ T cells and $CD8^+$ T cells within the tumor microenvironment induced by aAGd-NWs+RT. Radiation alone had a minimal impact on $CD4^+$ T-cell infiltration (2.68%) and $CD8^+$ T-cell infiltration (0.39%). In contrast, aAGd-NWs+RT significantly increased the proportions of infiltrating $CD4^+$ T cells (3.72%) and $CD8^+$ T cells (1.38%) (Fig. 8h, i and Supplementary Fig. 17a, b). Additionally, immunohistochemical staining of tumor tissues confirmed that the combination treatment of aAGd-NWs+RT substantially improved the infiltration of both $CD4^+$ and $CD8^+$ T cells (Supplementary Fig. 17c−e). Cytotoxic $CD8^+$ T lymphocytes (CTLs) play a crucial role in immunotherapy. We immunologically depleted $CD8^+$ T cells with anti-CD8a antibodies (αCD8a) to elucidate the importance of $CD8^+$ T cells in aAGd-NWs sensitized radiation therapy. The depletion of $CD8^+$ T cells significantly weakened

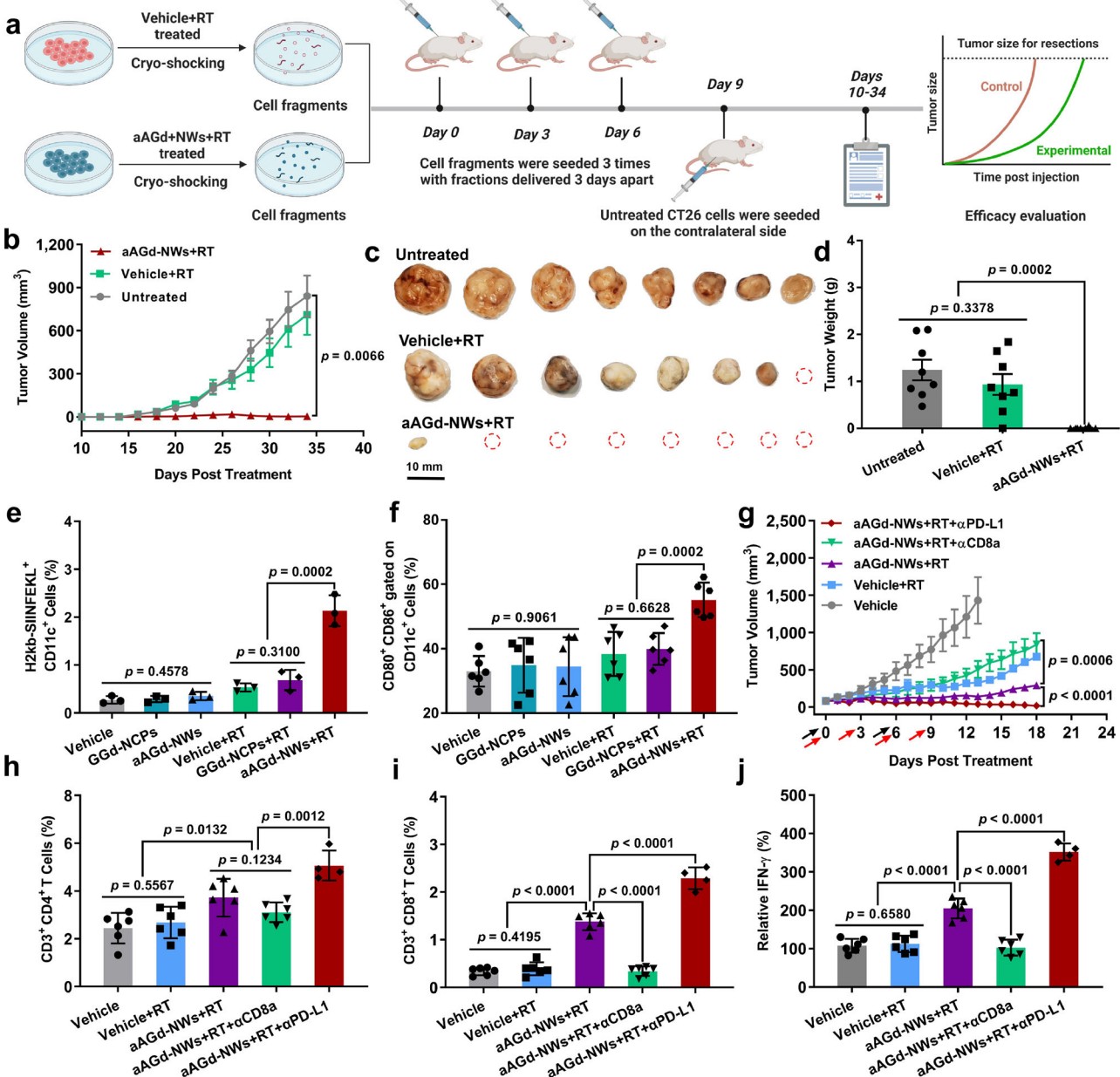

**Fig. 8 | Systemic antitumor immunity induced by aAGd-NW-sensitized radiation. a** Schematic illustration of aAGd-NWs sensitized radiation-mediated in situ vaccine. Created with BioRender.com. **b–d** Tumor growth curves (**b**), isolated tumor photographs (**c**), and tumor weights (**d**) of BALB/c mice immunized with RT and aAGd-NWs+RT treated tumor cells, respectively (*n* = 8 mice). **e** Flow cytometry analysis of CD11c⁺ and H2Kb-SIINFEKL⁺ DCs in TDLNs after Vehicle, GGd-NCPs, aAGd-NWs, Vehicle+RT, GGd-NCPs+RT, aAGd-NWs+RT treatments (*n* = 3 mice). **f** Flow cytometry analysis of DCs maturation (CD80⁺ and CD86⁺ gated within CD11c⁺) in TDLNs (*n* = 8 mice). **g** Tumor growth curves of CT26 colorectal tumor-bearing mice (*n* = 8 mice). Black and red arrows indicated drug/radiation and anti-CD8a/anti-PD-L1 treatments, respectively. RT (5 Gy × 2) was performed on day 0

and 6, respectively. Anti-Cd8a/anti-PD-L1 antibodies (10.0 mg kg⁻¹ × 4 with fractions delivered 3 days apart) were administered via intraperitoneal injection 6 h after X-ray irradiation. **h, i** Percentages of CD4⁺ (**h**) and CD8⁺ (**i**) T cells infiltrating within tumor tissues as detected by flow cytometry (*n* = 4 mice for the combinational group and *n* = 6 mice for other groups). **j** Relative multiples of IFN-γ secretion in the tumor tissues as detected by ELISA kit (*n* = 4 mice for the combinational group and *n* = 6 mice for other groups). All data were shown as mean ± SD. Two-tailed Student's *t*-test was used to calculate statistical differences between two groups, and one-way ANOVA analysis of variance was used for multiple groups. *p* values > 0.05 represented nonsignificance (N.S.) and *p* values < 0.05 represented statistically significant. Source data are provided as a Source Data file.

the antitumor activities of aAGd-NWs+RT. Conversely, immune checkpoint blockade therapies (αPD-1 or αPD-L1) targeting PD-1 or PD-L1 alleviate the PD-L1/PD-1-induced exhaustion of CD8⁺ T cells. Treatment with aAGd-NWs+RT in combination with αPD-L1 exhibited excellent synergy, resulting in 96.0% tumor growth inhibition (TGI), which was significantly higher than that achieved by aAGd-NWs+RT (84.6% TGI) (Fig. 8g, i). Thus, these findings clearly demonstrated the

involvement of CD8⁺ T cells in aAGd-NW + RT-induced antitumor immunity. We then evaluated the secretion of IFN-γ within tumor tissues. The aAGd-NWs+RT group exhibited enhanced IFN-γ secretion, which was further potentiated by αPD-L1 (Fig. 8j). Furthermore, the stable body weight observed in all treated mice further confirmed the biosafety of the combined therapeutic strategies in vivo (Supplementary Fig. 17f).

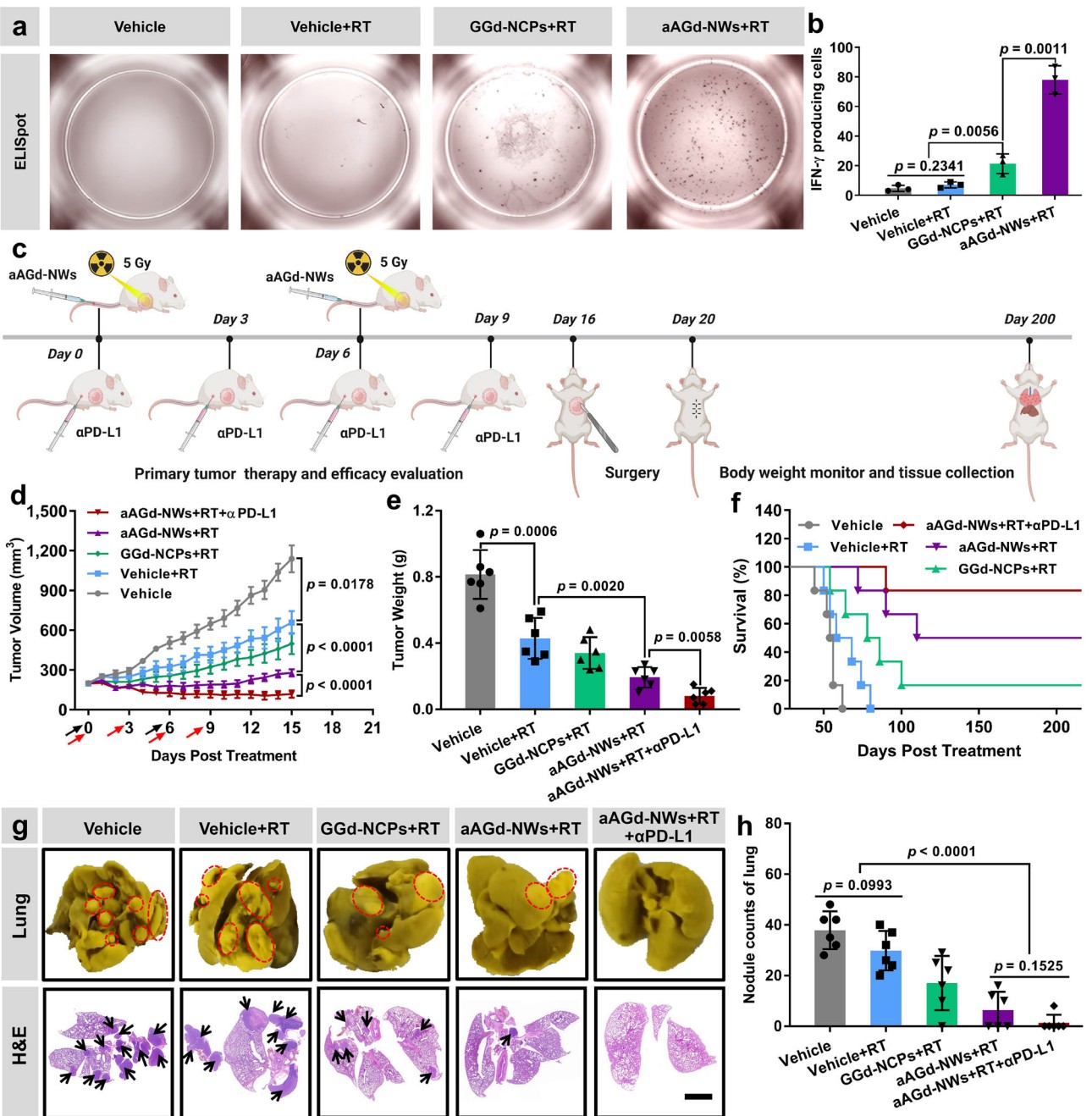

**Fig. 9 | Therapeutics of 4T1 metastatic breast cancer. a** The representative images of antigen-specific T Cells by IFN-γ ELISpot ($n = 3$ experimental repeats). This experiment was repeated twice independently with similar results. **b** The quantification of antigen-specific T Cells by IFN-γ ELISpot ($n = 3$ experimental repeats). **c** Schematic illustration of tumor therapeutic profiles. Created with BioRender.com. **d** Primary tumor growth curves of 4T1 breast tumor-bearing mice treated with Vehicle, Vehicle+RT, GGd-NCPs+RT, aAGd-NWs+RT and aAGd-NWs +RT + αPD-L1 ($n = 6$ mice). Black and red arrows indicated drug/radiation and anti-PD-L1 antibody treatments, respectively. RT 5 Gy × 2 with fractions delivered 6 days apart. Anti-PD-L1 antibody (10.0 mg kg⁻¹ × 4 with fractions delivered 3 days apart) were administered via intraperitoneal injection 6 h after radiation therapy. **e** Primary excised 4T1 breast tumor weights ($n = 6$ mice). **f** Survival curves of 4T1 breast tumor-bearing mice. **g** Images of lungs fixed by Bouin's solution and H&E sections of lungs, scale bar = 1 mm. **h** Quantification of metastatic lesions of the lungs ($n = 6$ mice). All data were shown as mean ± SD. Two-tailed Student's $t$-test was used to calculate statistical differences between two groups, and one-way ANOVA analysis of variance was used for multiple groups. $p$ values > 0.05 represented nonsignificance (N.S.) and $p$ values < 0.05 represented statistically significant. Source data are provided as a Source Data file.

## Anti-tumor immunotherapy

For the purpose of identifying OVA$_{257-264}$ antigen-specific T cell activation, we extracted cells from the spleens of B16-OVA tumor-bearing mice that had been immunized with different treatments. These cells were then incubated with the OVA$_{257-264}$ peptide (SIINFEKL) on polyvinylidene fluoride (PVDF)-lined microplates precoated with a capture antibody to IFN-γ, and cytokine spots are identified using an IFN-γ-specific detection antibody and an enzyme-linked conjugate. As depicted in the Fig. 9a, b, the number of OVA$_{257-264}$ peptide-specific CD8⁺ T cells in the splenocytes of mice treated by aAGd-NW + RT was significantly higher than those in the RT alone or GGd-NCPs+RT groups. These findings further substantiate that the combination of aAGd-NW and RT is exceptionally effective in inducing in situ vaccination and initiating antigen-specific anti-tumor immune responses.

Since aAGd-NWs have been demonstrated to induce in situ vaccination and prime CD8[+] T-cell-dependent antitumor immune responses, we further evaluated their capacity to inhibit metastasis in a spontaneous 4T1 metastatic breast cancer model (Fig. 9c). Compared to GGd-NCPs+RT or RT alone, aAGd-NWs significantly elevated the therapeutic effects of radiation on 4T1 primary tumors (Fig. 9d, e and Supplementary Fig. 18a). Furthermore, the stable body weights observed in all treated mice further confirmed the biosafety of the combined strategies in vivo, specifically the aAGd-NWs+RT treatment (Supplementary Fig. 18b). After 15 days of treatment, the 4T1 primary tumors and the surrounding skin tissues were surgically removed and carefully sutured. Due to the impact of metastasis on normal organs, the weights of the treated mice were monitored during a 200-day observation period and showed a significant reduction prior to death (Supplementary Fig. 18c). Notably, the 200-day long-term survival rate of the aAGd-NWs+RT group reached 50.0%, which was significantly higher than that of the vehicle (0%), RT (0%) and GGd-NCPs+RT (16.7%) groups. Furthermore, after potentiation with αPD-L1 treatment, aAGd-NWs+RT further increased the long-term survival rate to 83.5% (5/6) (Fig. 9f). Importantly, aAGd-NWs+RT + αPD-L1 treatment resulted in the fewest metastatic lesions within lung tissues, as confirmed by hematoxylin and eosin (H&E)-stained lung sections (Fig. 9g, h). These findings demonstrated the good ability of aAGd-NWs to sensitize radiation therapy and induce potent in situ vaccination, thereby potentiating the efficacy of αPD-L1 treatment against both primary and metastatic tumors.

## Discussion

Improving the efficacy of in situ vaccination has been demonstrated as an effective approach to stimulate adaptive antitumor immune responses, thereby enhancing the abscopal effects of radiation therapy (RT)[1–5]. However, the therapeutic effects of X-ray irradiation have been hindered by the weak absorption of X-rays in tumor tissues, primarily due to the low atomic number of their constituent elements[6]. In both our studies and those of others, doses of 4–10 Gy are commonly utilized to induce immunogenic death of tumor cells and initiate antitumor immunity[4–7]. Theoretically, RT has the potential to induce CRT translocation and damage DNA, thus promoting the activation of the cGAS/STING pathway in surrounding immune cells[21–23]. However, the efficiency of RT in inducing the exposure or release of these DAMPs is notably limited, leading to seemingly inadequate capacity to initiate anti-tumor immune responses[15–17]. The efficacy of RT-induced in situ vaccination is deemed unsatisfactory, evidenced by abscopal effects manifesting in less than 1% of clinical cases[8]. Consequently, there is a necessity for improved strategies to enhance RT-induced in situ vaccination. Recently, the approval of NBTXR3 (HfO$_2$ nanoparticles) in Europe has provided a means to sensitize RT by enhancing X-ray deposition specifically for the treatment of soft tissue sarcoma. Moreover, NBTXR3 has shown potential in potentiating anti-PD-1 treatment in various clinical trials. Consequently, NBTXR3-mediated radiosensitization holds promise for improving in situ tumor vaccination and potentially enhancing the efficacy of anti-PD-1 therapy in nonresponders[9]. Nevertheless, both preclinical investigations of NBTXR3 and our own studies have revealed that relying solely on the high-Z radiosensitization strategy yields only modest to moderate therapeutic outcomes[13,14]. Therefore, additional considerations should be given to mechanisms that prevent the induction of ICD and subsequent in situ vaccination.

To safeguard the integrity of their genomic structure, tumor cells swiftly activate multiple DNA repair pathways in response to X-ray-induced DNA damage[19]. To address this challenge, we successfully synthesized ultrafine chiral nanowires (aAGd-NWs) comprising the clinical antiviral drug vidarabine monophosphate (ara-AMP) and the high-Z element gadolinium (Gd) via supramolecular self-assembly, independent of any template usage. We utilized cryogenic transmission electron microscopy to analyze the structure of the ultrafine nanowires and unravel the drug molecules' self-assembly mechanism via theoretical simulation. Additionally, aAGd-NWs efficiently facilitated X-ray deposition, thereby boosting the generation of hydroxyl radicals (•OH). Then, after accumulation and deep penetration, ara-AMP could be gradually released from aAGd-NWs, which underwent phosphorylation to form vidarabine triphosphate (ara-ATP), which competitively hindered the repair processes of damaged DNA to synergize with the high-Z radiosensitization treatment. Consequently, tumor cells treated with aAGd-NWs in combination with RT exhibited increased release of damaged DNA fragments into the cytoplasm or extracellular space. These fragments were subsequently internalized by neighboring immune cells, triggering the activation of the cGAS-STING pathway[40–43]. Activation of this pathway further facilitates the secretion of type I interferons[40–43]. In addition to type I IFNs, our experimental findings also demonstrated that aAGd-NWs combined with RT synergistically induce CRT exposure, HMGB1 release, ATP secretion and other DAMPs (Fig. 7). CRT exposure and type I IFN secretion are crucial events for antigen cross-presentation. First, tumor cells with exposed CRT on their surface bind to CD91 receptors expressed on APCs, serving as an "eat me" signal that promotes the phagocytosis of tumor antigens. Recent studies have revealed that antigens phagocytosed via the CD91 pathway undergo degradation and cross-presentation to major histocompatibility complex class I (MHC-I) for CD8[+] T-cell priming[45–47]. Furthermore, type I IFNs have been reported to play a critical role in cross-presentation by impairing the acidification of antigen-presenting cell phagosomes, thus preventing the efficient digestion of exogenous antigens and facilitating their escape into the cytoplasm[18].

Recently, ara-AMP has gained widespread attention for the treatment of viral infections. However, it was initially designed as an anticancer drug, targeting various enzymes involved in DNA synthesis or repair[30–32]. Unfortunately, free ara-AMP has inherent limitations, such as a short half-life and low tumor accumulation capacity, which compromise its efficacy in tumor treatment[30–32]. To overcome these challenges, we developed ultrafine nanowires, referred to as aAGd-NWs, through the self-assembly of ara-AMP and Gd. This self-assembly not only improved the biocompatibility and pharmacokinetic behavior of the components and prevented rapid elimination but also facilitated their accumulation and deep penetration into tumor tissues, leading to superior therapeutic effects (Figs. 5 and 6). However, it is important to consider the potential release of Gd, which can cause substantial damage to normal tissues, including the kidneys. Considering that aAGd-NWs release drugs in a pH-dependent manner, we hypothesize that the coordination state of Gd in aAGd-NWs was closely linked to the pKa of ara-AMP (pKa$_1$ = 3.8, pKa$_2$ = 6.2) (Supplementary Fig. 19a)[48]. Theoretically, when pH > 6.2, aAGd-NWs maintain a relatively stable particulate state. As the pH value gradually decreased, the phosphate partially became mono-protonated (3.8 < pH ≤ 6.2), these nanowires would still maintain their particulate or coordination state. When pH ≤ 3.8, free Gd$^{3+}$ is hypothesized to be progressively released from aAGd-NWs owing to the further protonation of phosphate groups (Supplementary Fig. 19b). However, it is critical to emphasize that such a low pH environment represents a virtually unattainable condition within human metabolism. As shown in the Supplementary Fig. 19c, aAGd-NWs exhibited robust stability under physiological condition (pH = 7.4). With the pH value decreasing (pH = 5.0), aAGd-NWs will gradually degrade, which was confirmed by HRTEM (Supplementary Fig. 19d). Preliminary studies from our group also have demonstrated that gadolinium and 5'-GMP ligands create nanoscale coordination polymers, which were predominantly excreted in the urine in coordinated forms, instead of as free agents[13].

To further assess the acute toxicity of aAGd-NWs and free GdCl$_3$ in healthy BALB/c mice, we conducted further evaluations. The mice were independently divided into three groups (n = 3): saline, free GdCl$_3$

([Gd] = 3.0 mg kg$^{-1}$ × 4) and aAGd-NWs ([Gd] = 15.7 mg kg$^{-1}$ × 4, 5.2 times the dose of GdCl$_3$). Intravenous injections were administered every two days for four cycles, and then the mice were sacrificed. As shown in Supplementary Fig. 20a, mice in the free GdCl$_3$ group exhibited significant weight loss, whereas those in the saline and aAGd-NWs groups did not. Serum biochemical analysis revealed severe kidney damage in the free GdCl$_3$ group, whereas there was no statistically significant difference between the saline and aAGd-NWs groups (Supplementary Fig. 20b, c). Histological examination of renal specimens revealed lesions in only the free GdCl$_3$ group, characterized by multifocal chronic inflammation and interstitial edema (Supplementary Fig. 20d), indicating the relative biosafety of aAGd-NWs. Unlike the use of contrast agents in the general population, aAGd-NWs are specifically targeted for cancer treatment, often allowing for a broader tolerance in terms of safety. Typically, the clinical dosage of gadolinium-based contrast agents (e.g., Magnevist) is approximately 0.1 mmol kg$^{-1}$, which has been proven to have a good safety profile. Extrapolating based on body surface area[49], the dosing range of Gd for mice is about 1.0 - 1.2 mmol kg$^{-1}$, comparable to the dosage of gadolinium-based contrast agents used in mice for MRI purposes[50,51]. The cumulative dose of Gd (0.1 mmol kg$^{-1}$ × 2) we employed is obviously lower than the single dose for Gd-based contrast agents. Results from acute toxicity studies further demonstrate that even when the total dose was increased from 0.1 mmol kg$^{-1}$ × 2 to 0.1 mmol kg$^{-1}$ × 4, aAGd-NWs continued to exhibit excellent in vivo safety (Supplementary Fig. 20). Additionally, serum biochemical analysis of parameters such as alanine aminotransferase (ALT), aspartate aminotransferase (AST), creatinine (CREA), and urinary anine (UREA) indicates that while free ara-AMP may cause some liver and kidney function impairment, there is no significant statistical difference between the effects of aAGd-NWs and those of Saline, as evidenced in Supplementary Fig. 21a–d. Furthermore, histological examinations (H&E staining) of major organs reveal that the impact of aAGd-NWs is not significantly different from that of the Saline group (Supplementary Fig. 22).

Besides, we also employed various methods and conducted experiments to pursue the synthesis of left-handed chirality aAGd-NWs. Despite extensive efforts, this goal has not yet been achieved. As demonstrated in Supplementary Fig. 23a, modifications the self-assembly ratio of ara-AMP and Gd, and adjustments in the reaction temperature, appear to have no impact on the circular dichroism (CD) values of ara-AMP-Gd NCPs. Conversely, substituting the ligand molecule from ara-AMP to adenosine monophosphate (AMP), they self-assemble to form spherical AMP-Gd NCPs[52] (AGd-NCPs, Supplementary Fig. 23b) and led to a notable reversal in the CD values, irrespective of the self-assembly ratios and reaction temperatures. These findings indicate that the chirality of NCPs is predominantly influenced by the ligand molecules, rather than changes in external conditions. The key distinction between AMP and ara-AMP lies in the spatial isomerism of the 3′-OH group. The simulation results revealed that the presence of isomerized 3′-OH in ara-AMP restricts the free rotation of the adenine group (Supplementary Fig. 24). In the case of AMP coordination, the C-N bond connecting the adenine group and ribose group in AMP molecules exhibits unrestricted rotation, leading to the formation of an anisotropic amorphous structure that subsequently assembles into spherical particles. Conversely, due to the steric hindrance imposed by the 3′-OH group, the C-N bond between the adenine group and ribose group in ara-AMP molecules lacks free rotation, maintaining a rigid conformation throughout the assembly process, which further undergoes cooperative stacking to generate nanowires.

In conclusion, we successfully engineered chiral and ultrafine aAGd-NWs through coordination-driven self-assembly of the clinically available ara-AMP and high-Z element Gd. These aAGd-NWs demonstrated the ability to sensitize radiation therapy (RT), eliciting potent ICD and functioning as an in situ vaccine. Moreover, they synergistically enhanced the efficacy of immune checkpoint blockade therapy against both primary and metastatic tumors. The well-established aAGd-NWs, characterized by their good therapeutic capacity and biocompatibility, offer a perspective for the field of radioimmunotherapy.

## Methods

### Ethical statement and confirmation

The protocol for animal experiments in this study underwent rigorous review and approval by the Animal Ethical and Welfare Committee of Nanjing University (Approval No. IACUC-D2202156) and was conducted in strict accordance with the principles outlined by the Association for Assessment and Accreditation of Laboratory Animal Care International (AAALAC). All animal testing was carried out in full compliance with applicable local and national ethical regulations.

### Materials

Gadolinium (III) chloride hexahydrate was purchased from Energy Chemical (China). Vidarabine monophosphate (ara-AMP) was purchased from Adamas-Beta Biological Technology Co., Ltd (China). Lyso-tracker Green, crystal violet 4′,6′-diamidino-2-phenylindole (DAPI) and 2′,7′-Dichlorodihydrofluorescein diacetate (H$_2$DCFDA) were purchased from Sigma-Aldrich (USA). Cell counting kit-8 (CCK-8) was obtained from Dojindo Laboratories (Japan). PicoGreen dsDNA quantitation reagent was purchased from Yeasen Biological Technology Co., Ltd (China). Mouse HMGB1, IFN-γ, and IFN-β precoated ELISA Kit was obtained from Dakewe Biotech Co., Ltd. Anti-gamma H2Aχ (phospho S139) antibody [9F3] (Cat# ab26350, diluted 1:200 with 3% BSA), TUNEL Assay Kit - BrdU-Red (Cat# ab66110, diluted 1:200 with 3% BSA), Anti-Ki67 antibody (Cat# ab15580, diluted 1:500 with 3% BSA), Anti-Calreticulin-ER Marker (Alexa Fluor® 488) (Cat# ab196159, diluted 1:500 with 3% BSA), Anti-beta Actin antibody [mAbcam 8226] - Loading Control (Cat# ab8226), Recombinant Anti-STING antibody [EPR25090-107] (Cat# ab288157, diluted 1:500 with 3% BSA) and Recombinant Anti-IRF3 antibody [EPR2418Y] (Cat# ab68481, diluted 1:500 with 3% BSA) were supplied by Abcam (USA). Phospho-STING (Ser366) Polyclonal Antibody (Cat# PA5-105674, diluted 1:500 with 3% BSA) and Phospho-IRF3 (Ser386) Polyclonal Antibody (Cat# PA5-121307, diluted 1:500 with 3% BSA) were purchased from Thermofisher (USA). HRP conjugated Goat Anti-Rabbit IgG (H + L) (Cat# GB23303, diluted 1:300 with 3% BSA) was obtained from Servicebio (China). APC anti-mouse CD80 Antibody [16-10A1] (Cat# 104713, 1.0 μg per million cells in 100 μL volume), PE anti-mouse CD86 Antibody [GL-1] (Cat# 105007, 0.25 μg per million cells in 100 μL volume), FITC anti-mouse CD11c Antibody [N418] (Cat# 117306, 0.25 μg per million cells in 100 μL volume), APC anti-mouse CD3 Antibody [17A2] (Cat# 100236, 0.5 μg per million cells in 100 μL volume), PE anti-mouse CD4 Antibody [GK1.5] (Cat# 100408, 0.25 μg per million cells in 100 μL volume), FITC anti-mouse CD8a Antibody [53-6.7] (Cat# 100706, 1.0 μg per million cells in 100 μL volume), APC anti-mouse H-2Kb Antibody [AF6-88.5] (Cat# 116506, 1.0 μg per million cells in 100 μL volume), Purified anti-mouse IFN-γ Antibody [R4-6A2] (Cat# 505702, 2.0 μg mL$^{-1}$) and Ultra-LEAF™ Purified anti-mouse CD8a Antibody [53−6.7] (Cat# 100764, 10.0 mg kg$^{-1}$) were purchased from BioLegend (USA). In Vivo MAb anti-mouse PD-L1(B7-H1) [Clone: 10 F.9G2] (Cat# BE0101, 10.0 mg kg$^{-1}$) was purchased from BioXcell (USA).

### Instruments

The morphologies, crystal structures and elemental composition were characterized using transmission electron microscopy (TEM) (Hitachi HT7700, Japan) and high-resolution TEM (HRTEM) combined energy dispersive spectroscopy (EDS) (JEM-2100F, JEOL, Japan). Cryo-Transmission Electron Microscope (cryo-TEM) images were acquired by the Thermo Scientific Glacios 2 cryo-EM (USA). Powder X-ray diffraction (PXRD) patterns were obtained using a D8 ADVANCE XRD

(Bruker, Germany) with Cu Kα radiation (λ = 1.54 Å). X-ray photoelectron spectroscopy (XPS) spectra were measured using a PHI 5000 VersaProbe (Ulvac-Phi, Japan) with Al Kα radiation (hν = 1486.6 eV). Ultraviolet-visible (UV-vis) absorption spectra were recorded with a Shimadzu UV-vis spectrophotometer (UV3600, Japan). Fourier transform infrared (FT-IR) spectra were recorded with a Fourier-Transform Infrared Spectrometer (Bruker, Vertex 80 v and Tensor 27, Germany). Electron Spin Resonance (ESR) specta were tested by the EMS Plus ESR Spectrometer (Bruker, Germany). The amount of $Gd^{3+}$ was confirmed by an Inductively Coupled Plasma Optical Emission Spectrometer (ICP-OES) (Perkin Elmer Optima 5300 DV, USA). Flow cytometry for typing of all immune cells (BD, FACSCalibur, USA). The X-ray irradiation equipment is the Rad Source RS2000 X-ray irradiator (Model: X-ray Irradiator RS2000, USA). The instrument's technical specifications comprise a dose rate of 10 Gy min$^{-1}$ for cells and 1.2 Gy min$^{-1}$ for small animals. The irradiation plane is situated 25 cm away in the cone-shaped irradiation field. The operating specifications of the power supply are 220 volts, 60 Hz, 40 amps.

## Cell lines

The mouse CT26 (Cat# CRL-2638), RAW264.7 (Cat# SC-6005), B16-OVA (Cat# CRL-6323) and 4T1 cell lines (Cat# CRL-2539) are purchased and authenticated by the American Type Culture Collection (ATCC). All cell lines in this study get tested without mycoplasma contamination.

## Animals

BALB/c mice were purchased from medicine center of Yangzhou university (Yangzhou, China). All animal work was approved by the Institution Animal Care and Use Committee of Nanjing University (IACUC-D2202156) and conducted in accordance with the principles of the Association for Assessment and Accreditation of Laboratory Animal Care International (AAALAC). All animals were housed with the light cycle of 12 h: 12 h, ambient temperature at 22 degree Celsius, and relative humiditrange between 40–70%. The maximum tumor size allowed by the ethics committee or institutional review board was 2000 mm$^3$, and we confirmed that the maximum tumor burden would not be exceeded in all experimental groups.

## Preparation and characterization of aAGd-NWs and GGd-NCPs

The aqueous solution of 10 mM ara-AMP (20.0 mL) was instilled into 100 mL flask with 10 mM solution of $GdCl_3$ (20.0 mL), and stirred for 1 h at room temperature. After 1 h of coordination, the precipitates were collected by centrifugation (3000 g × 30 min). The obtained precipitate was washed with distilled water three times. Next, these precipitates were dispersed in the 20.0 mL of aqueous solution (containing 2% mouse serum albumin (MSA) as emulsifier and aggregation inhibitor) by ultrasound (650 W, 45% intensity × 8 min) to obtain nanoscale coordination ultrafine aAGd-NWs. On the other hand, ara-AMP was replaced with 5′-GMP, and coordinated with Gd to obtain spherical GGd-NCPs in similar ways.

## Characterization of aAGd-NWs

The particle size of aAGd-NWs was determined by dynamic light scattering (DLS, Brookhaven 90 plus Zeta). Then, aAGd-NWs (final concentration of 10 μM) were uniformly dispersed in 2.0 mL of deionized $H_2O$ (25 °C), and then added to the cuvette and inserted with AQ-961 palladium electrode for Zeta Potential detection (Brookhaven 90 plus Zeta).

## Preparation of ICG@aAGd-NWs

10 mM ara-AMP (20.0 mL) and 1.0 mM ICG (0.1 mL) were mixed and then slowly added with 10 mM $GdCl_3$ (20.0 mL) dropwise and stirred for 1 h. After coordination, the precipitates were washed and collected by centrifugation (3000 g × 30 min). Then the obtained precipitates were dispersed into ICG@aAGd-NWs by the same method.

## Assessment of hydroxyl radical generation in vitro

We used UV colorimetric method to detect the generation of hydroxyl radicals (·OH) under X-ray irradiation based on the decay of methylene blue (MB). Briefly, Vehicle, $GdCl_3$, GGd-NCPs or aAGd-NWs were mixed with MB solutions ([Gd] = 20 μM and [MB] = 15 μg mL$^{-1}$). After irradiating with different doses (0, 5 Gy × 1, 5 Gy × 2, 5 Gy × 3, 5 Gy × 4), the absorption of MB at 664 nm was detected to observe the degradation of MB, respectively.

## Cumulative ara-AMP release

The release of ara-AMP from aAGd-NWs was studied at different pH values. aAGd-NWs ([aAGd-NWs] = 1 mM × 1 mL) was packed into Solarbio (10 kD) dialysis bags and stirred in 10.0 mL 1 × HEPES buffers containing 10% serum (37 °C) at pH = 7.4, 6.5, 5.5, respectively. 100.0 μL of the sample was collected from the dialysates at 0, 2, 4, 8, 12, 24, 48 and 72 h, respectively. ara-AMP concentration was determined by reverse-phase high performance liquid chromatography (RP-HPLC, LC-20AT, Shimadzu, Kyoto, Japan) method, and Gemini 5 μm C18 (4.6 × 250 mm) column was used. Mobile phase was a mixture 20:80 (v/v) of $CH_3OH$ and an aqueous solution (10 mM $K_2HPO_4$ and 7 mM tetrabutyl ammonium hydrogen sulphate, adjusted at pH 7.0 by NaOH), UV-Vis detector wavelength was 258 nm. The flow rate of the mobile phase was 1 mL min$^{-1}$ and the pool temperature was set as 25 °C[53].

## Intracellular uptake behavior analysis

CT26 tumor cells were seeded in a glass bottom cell culture dish (NEST, 20 mm) at a density of $2 \times 10^5$ cells per dish. The cells were incubated with ICG@aAGd-NWs ([Gd] = 50 μM, [ara-AMP] = 50 μM; $E_x$ = 789 nm, $E_m$ = 813 nm) for 6 h. LysoTracker Green ($E_x$ = 504 nm, $E_m$ = 511 nm) and tumor cells were co-incubated for 30 min at 37 °C, washed twice with PBS, and then stained with DAPI ($E_x$ = 364 nm, $E_m$ = 454 nm). Then, tumor cells were washed 3 times with PBS and observed via Olympus FV3000 CLSM. Turning on different lasers in CLSM sequentially and separately to prevent crosstalk. Then, ImageJ (Colocalization Finder) was used for the calculation of the Pearson colocalization coefficient (PCC).

## Intracellular ROS generation

CT26 cells were seeded in a 96-well plate at a density of 8000 cells per well and attached overnight. The cells were incubated with Vehicle, GGd-NCPs, aAGd-NWs ([Gd] = 50 μM, [ara-AMP] = 50 μM) for 6 h. Before X-ray irradiation, $H_2DCFDA$ (1:1000, used as ROS probe to detect intracellular ROS generation) was incubated for 1 h at 37 °C and washed with PBS for three times. After irradiation (0 Gy or 5 Gy × 1), DAPI was used to stain the nucleus. Subsequently, immunofluorescence images were obtained from Olympus FV3000 CLSM and analyzed by ImageJ Software.

## Detection of intracellular γ-H2Aχ

CT26 cells were seeded in confocal dishes at $2 \times 10^5$ per dish and incubated overnight for attachment, which were divided into six groups of Vehicle, Vehicle+RT, Gd-NCPs, GGd-NCPs+RT, aAGd-NWs, aAGd-NWs+RT ([Gd] = 50 μM, [ara-AMP] = 50 μM). After incubation for 6 h, the cells were treated with radiation (0 Gy or 5 Gy × 1). Two hours after irradiation, all treatments were removed, and tumor cells were fixed with 4% paraformaldehyde and washed with PBS. Then, 0.3% Triton-X was used to perforate the nucleus. Then, cells were added with 1% bovine serum albumin solution as a blocking buffer, and incubated with γ-H2Aχ mouse monoclonal primary antibody (1:500, Abcam, UK) for 1 h and washed with PBS. After that, Alexa Fluor 488

conjugated secondary antibody (1:500, Bioss, China) was added and incubated for 1 h. Then, DAPI was used to stain the nucleus. Tumor cells were observed under Olympus FV3000 CLSM, and analyzed by ImageJ software.

## Cytotoxicity

CT26 cells were cultured in Dulbecco's Modified Eagle's Medium (DMEM, Gibco) supplemented with 10% fetal bovine serum (FBS, Gibco), 100 μg mL$^{-1}$ penicillin, 100 μg mL$^{-1}$ streptomycin and 50 μg mL$^{-1}$ gentamicin and grown in a humidified atmosphere with 5% $CO_2$ at 37 °C. To study the cytotoxicity of aAGd-NWs and GGd-NCPs with or without X-ray irradiation, CT26 cells were seeded in 96-well plates with 8000 cells per well. After attachment, aAGd-NWs and GGd-NCPs ([Gd] = 0, 3.12, 6.25, 12.5, 25.0 and 50 μM) were added and incubated for 24 h. After X-ray irradiation (5 Gy × 1), CT26 cells were cultured for another 24 h. Then the cell viability was determined by CCK-8 (Dojindo, Japan) assay. Measure the absorbance at 450 nm using a microplate reader.

## aAGd-NWs deeply penetrated 3D tumor spheroids

To evaluate the 3D Spheroid penetration, CT26 cells (3 × 10$^3$ per well) were seeded into ultralow attachment 96-well plates (Corning, 7007, USA). Indocyanine green (ICG) was integrated into GGd-NCPs and aAGd-NWs to obtain ICG@GGd-NCPs and ICG@aAGd-NWs, respectively. Then, CT26 3D tumor spheroids were co-incubated with ICG@GGd-NCPs and ICG@aAGd-NWs for 8 h. Then the red fluorescence of ICG within 3D spheroids was recorded by Olympus FV3000 CLSM. To delve deeper into the penetration capabilities of ultrafine aAGd-NWs in 3D CT26 spheroids, these spheroids were dissociated into single cells for flow cytometry analysis. The dissociated single-cell suspension was subjected to three washes with PBS and was then collected by short-duration and low-speed centrifugation (2000g × 5 min) to remove unuptaked ICG@aAGd-NWs. Ultimately, the cells treated with ICG@aAGd-NWs were resuspended in PBS for flow cytometry analysis. To test the cytotoxicity of aAGd-NW-sensitized radiation within CT26 3D tumor spheroids, CT26 3D spheroids were incubated with Vehicle, GGd-NCPs ([Gd] = 50 μM), aAGd-NWs ([Gd] = 50 μM, [ara-AMP] = 50 μM) for 12 h and then irradiated (0 or 5 Gy × 1). After 24 h, calcein-AM and propidium iodide (PI) were used to stain live and dead cells and the sizes of 3D tumor spheroids after various treatments were measured by Nikon Eclipse Ti (Japan) at days 5 and 10, respectively.

## Biodistribution studies and pharmacokinetics of aAGd-NWs in vivo

To evaluate the in vivo pharmacokinetics of aAGd-NWs, CT26 tumor-bearing mice (100–150 mm$^3$, $n$ = 3) were intravenously injected with free ara-AMP or aAd-NWs ([Gd] = 15.7 mg kg$^{-1}$, [ara-AMP] = 34.7 mg kg$^{-1}$), and blood, tumor tissues were collected from mice at different time points (0.5, 1, 3, 6, 12, 24, 48 h) respectively. Then, methanol ($CH_3OH$) and 0.01 M HCl were added into the collected blood to extract ara-AMP. Then, ara-AMP concentration was determined by reverse-phase high performance liquid chromatography (RP-HPLC, LC-20AT, Shimadzu, Kyoto, Japan) method, and Gemini 5 μm C18 (4.6 × 250 mm) column was used. Mobile phase was a mixture 20:80 (v/v) of $CH_3OH$ and an aqueous solution (10 mM $K_2HPO_4$ and 7 mM tetrabutyl ammonium hydrogen sulphate, adjusted at pH 7.0 by NaOH), UV-Vis detector wavelength was 258 nm. The flow rate of the mobile phase was 1 mL min$^{-1}$ and the pool temperature was set as 25 °C[54]. To assess the accumulation of Gd, tumor tissues were crushed and homogenized, followed by burning and nitrification, and then diluted with 1.0 mL of 1% $HNO_3$ to quantify [Gd] by ICP-OES.

## Radiosensitization of aAGd-NWs in CT26 tumor model

When the tumor volume reached 80–100 mm$^3$, the mice were administered with Vehicle (Saline containing 2% mouse serum albumin), GdCl$_3$ ([Gd] = 15.7 mg kg$^{-1}$), ara-AMP ([ara-AMP] = 34.7 mg kg$^{-1}$), aAGd-NWs ([Gd] = 15.7 mg kg$^{-1}$, [ara-AMP] = 34.7 mg kg$^{-1}$) with or without X-ray irradiation (5 Gy × 2). The treatments were given on day 0 and day 6, respectively. X-ray irradiation (5 Gy × 2, 6 days apart) was given 6 h after drug administration. Then, tumor size and body weights were monitored every day (tumor volume: V = width$^2$ × length / 2). Tumor tissues, hearts, livers, spleens, lungs, and kidneys were collected and fixed in 4% formalin, embedded in paraffin, sliced, and stained with Hematoxylin and Eosin (H&E).

## Immunofluorescence of γ-H2Aχ, TUNEL and IHC analysis of Ki67

The CT26 tumor-bearing mice with tumor volume of 150–200 mm$^3$ were classified: Vehicle, GGd-NCPs, aAGd-NWs ([Gd] = 15.7 mg kg$^{-1}$, [ara-AMP] = 34.7 mg kg$^{-1}$) with or without X-ray irradiation (5 Gy × 1). The corresponding drugs were administered intravenously 6 h before radiotherapy. Then, tumor tissues were collected for evaluation of γ-H2Aχ 24 h post irradiation. Tumor tissues were sliced and stained with γ-H2Aχ mouse monoclonal primary antibody (diluted 1:200 with 3% BSA, Abcam, UK) and secondary antibody conjugated to Alexa Fluor 488 (Bioss, China) to detect DNA double-strand breaks. According to the manufacturer's protocol, tumors were collected from six groups at 48 h post different treatments for terminal deoxynucleotidyl transferase-mediated dUTP-biotin nick end labeling (TUNEL) assay and Ki67 staining, respectively.

## Detection of cytosolic DNA damages

To detect cytosolic DNA damages, CT26 cells were seeded in a glass bottom cell culture dish (NEST, 20 mm) at a density of 2 × 10$^5$ cells per dish. The cells were treated with Vehicle, GGd-NCPs, aAGd-NWs with or without X-ray irradiation ([Gd] = 50 μM, [ara-AMP] = 50 μM) and incubated for 24 h. Then, PicoGreen staining was performed 6 h and 24 h post irradiation, respectively. For PicoGreen staining, cells were incubated with PicoGreen dsDNA Quantitation Reagent (diluted 1:200 with PBS, Yeasen, CHINA) for 10 min at 37 °C, washed with PBS and stained with DAPI (Beyotime, China). Fluorescence images were obtained from Olympus FV3000 CLSM and analyzed with ImageJ Software.

## cGAS-STING activation and IFN-β secretion

CT26 cells were cultured in a 6-well plate with a density of 5 × 10$^5$ cells per well for 24 h. Subsequently, after an additional 24 h of incubation, the cells were treated with aAGd-NWs (50 μM). Following this treatment, X-ray irradiation (5 Gy) was performed, and RAW264.7 cells were added for co-culture for an additional 24 h. To assess the activation of the cGAS-STING pathway, a mixture of non-denatured Tissue/Cell Lysate Kit (Solarbio) was used along with a broad-spectrum protease inhibitor cocktail (EDTA-free, BOSTER) and a broad-spectrum phosphatase inhibitor cocktail (EDTA-free, BOSTER) to extract proteins from the mixed cells. Finally, 30 μL of the isolated samples were used for Western blotting analysis to confirm the key proteins involved in the cGAS-STING pathway (p-STING, STING, p-IRF3, IRF3, and β-actin). Besides, 30 μL of the supernatant was used to detect the secretion of IFN-β by using IFN-β ELISA kit.

## CRT exposure detection

CT26 cells were seeded on glass bottom cell culture dish (NEST, 20 mm) with a density of 2 × 10$^5$ cells per dish. After 24 h, Vehicle or aAGd-NWs ([Gd] = 50 μM, [ara-AMP] = 50 μM) were added into culture medium and incubated for 24 h followed by X-ray irradiation (0 or 5 Gy × 1). After 4 hours incubation, cells were washed with PBS for three times and stained with CRT-antibody (diluted 1:500 with 1% BSA, Abcam Rb mAb to Calreticulin [EPR3924], UK) for 1 h. Subsequently, tumor cells were washed with PBS for three times and stained with Alexa Fluor 647 conjugated goat anti-rabbit IgG (H&L, diluted 1:200 with 1% BSA, Zen-Bioscience, China) for another 1 h at 37 °C. Then, tumor cells were stained with DAPI (Beyotime, China).

Immunofluorescence images were obtained from Olympus FV3000 CLSM and analyzed with ImageJ Software.

## Measurement of HMGB1 and ATP release

According to the manufacturer's protocol, HMGB1 concentration in the cytoplasm after different treatments were detected by ELISA kit (Yifeixue Bio, China). According to the manufacturer's protocol, ATP concentration in the cell supernatants after the specified treatments were measured by the ATP detection kit (Beyotime, China). Microplate reader (VICTOR® Nivo) was used to measure luminescence and absorbance, respectively.

## Prophylactic vaccination in vivo

The CT26 cells were treated with Vehicle+RT and AGd-NWs+RT ([Gd] = 15.7 mg kg$^{-1}$ and [ara-AMP] = 34.7 mg kg$^{-1}$) respectively. Four hours after irradiation (0 Gy or 5 Gy × 1), CT26 cells were collected and frozen in liquid nitrogen for 12 h. Subsequently, the cell fragments ($3 \times 10^5$ cells per mouse) were inoculated into the left abdomen of the healthy mice, and three consecutive immunizations were performed on days 0, 3, and 6, respectively. On day 9, mice were inoculated with untreated CT26 cells ($3 \times 10^5$ cells per mouse) on the contralateral abdomen, and tumor growth was recorded over the following period.

## Gating strategy and methodology of flow cytometry

All of the flow cytometry experiments were adopted with similar gating strategy. Forward Scatter (FSC) and Side Scatter (SSC) dot maps were established during the running process, and the voltage was adjusted to ensure that all events were visible on the dot maps. Gating was then performed to select events with appropriate FSC (200–600) and SSC (200–600) values, while events with low FSC/low SSC and low FSC/high SSC were excluded as they represented cell debris and air bubbles. Specific cell types were gated using fluorescently labeled antibodies, including DCs (CD80$^+$ and CD86$^+$ gated on CD11c$^+$), CD4$^+$ T cells (CD3$^+$ and CD4$^+$), CD8$^+$ T cells (CD3$^+$ and CD8$^+$), OVA-specific DCs (CD11c$^+$ H2Kb-SIINFEKL$^+$).

## Antigen presentation

To detect the impact of aAGd-NWs+RT on specific antigen presentation, B16-OVA subcutaneous tumor model was established on C57 mice. Initially, $5 \times 10^5$ B16-OVA cells were subcutaneously injected into the mice. After a period of 7 days, when the average tumor size in the mice carrying B16-OVA tumors reached approximately 100 mm$^3$, the mice were grouped randomly into six distinct categories: Vehicle, GGd-NCPs, aAGd-NWs, Vehicle+RT, GGd-NCPs+RT, and aAGd-NWs+RT ([Gd] = 15.7 mg kg$^{-1}$ and [ara-AMP] = 34.7 mg kg$^{-1}$). Treatments were performed on day 0 and day 6, respectively, and followed by RT (5 Gy × 2) 6 h post treatment. On day 8, tumor draining lymph nodes were harvested and processed into single-cell suspensions. The cells thus obtained were stained for 30 min using a combination of FITC-CD11c and APC-H2Kb-SIINFEKL antibodies (BioLegend, USA). Subsequently, these cells were analyzed through flow cytometry using the BD FACSCalibur system (USA).

## In vivo DC maturation

To detect DC maturation in tumor-draining lymph nodes (TDLNs), TDLNs were harvested 5 days after different treatments (Vehicle, Vehicle+RT, GGd-NCPs, GGd-NCPs+RT, aAGd-NWs and aAGd-NWs +RT). These collected TDLNs were digested and filtered through nylon mesh filters to obtain single cell suspensions. Subsequently, cells were stained with PE/Cyanine7 anti-mouse CD11c, PE anti-mouse CD86 and APC anti-mouse CD80 antibodies (BioLegend, America) for 30 min and then detected by flow cytometry (BD Calibur).

## Therapeutic effects of aAGd-NW-sensitized radiation therapy

CT26 cells ($5 \times 10^5$) were subcutaneously injected into the lower abdomen of male mice to establish the CT26 tumor model. Ten days after inoculation, CT26 tumor-bearing mice were randomly divided into 5 groups: Vehicle, Vehicle+RT, aAGd-NWs+RT, aAGd-NWs +RT+αCD8a and aAGd-NWs+RT + αPD-L1 ([Gd] = 15.7 mg kg$^{-1}$, ([ara-AMP] = 34.7 mg kg$^{-1}$), [αCD8a] = 10 mg kg$^{-1}$ and [αPD-L1] = 10 mg kg$^{-1}$). Treatments were performed on day 0 and day 6, respectively, and followed by radiotherapy (5 Gy × 2) 6 h later. 6 h post X-ray irradiation, mice were intraperitoneally injected with anti-PD-L1 antibody (10 mg kg$^{-1}$ × 4 with fractions delivered 3 days apart, Clone: 10 F.9G2, BioXcell, America) and anti-CD8a antibody (10 mg kg$^{-1}$ × 4 with fractions delivered 3 days apart, Clone: 53-6.7, BioLegend, America), respectively.

## IHC staining of CD8$^+$ T cell infiltration

Immunohistochemical staining was performed on tumor tissues on day 18 to detect the infiltration of CD8$^+$ T cells. Tumor tissues were embedded in paraffin, sectioned, and incubated with primary anti-CD8 antibody (Abcam, USA) overnight at 4 °C and then incubated with the HRP-conjugated secondary antibodies at 37 °C in dark for 1 h. Then, chromogenic reagents (Diaminobenzidine and Hematoxylin solution) were used to stain the CD8$^+$ T cells and cell nucleus, respectively. The immunochemical images were obtained from Nikon Eclipse Ti (Japan) and analyzed by ImageJ software. IFN-γ concentrations in the tumor tissues following the indicated treatments was measured by ELISA kit (BioLegend, USA), according to the manufacturer's protocol.

## IFN-γ ELISpot assay

To assess the effects of aAGd-NWs+RT in promoting antigen specific CD8$^+$ T cell activation, we established B16-OVA subcutaneous tumor model upon C57 mice. B16-OVA cells were subcutaneously injected with $5 \times 10^5$ cells. When the average tumor size in mice carrying B16-OVA reached 150–200 mm$^3$, they were randomly divided into four groups: Vehicle, Vehicle+RT, GGd-NCPs+RT, and aAGd-NWs+RT ([Gd] = 15.7 mg kg$^{-1}$ and [ara-AMP] = 34.7 mg kg$^{-1}$). Treatments were performed on day 0 and day 6, respectively, and followed by radiotherapy (5 Gy × 2) 6 h post treatments. On Day 15, spleens were harvested for IFN-γ ELISpot assay (Dakewe Biotech, 2210005). Spleens were grinded and erythrocyte-lysized single-cell suspensions were obtained by ACK lysis solution and filtering through a nylon mesh. For IFN-γ ELISpot assay, on the first day, the pre-coated plates were activated by coculturing with 200 μL of 1640 culture medium (supplemented with penicillin/streptomycin and 10% FBS) for 10 min. Then, culture medium was removed and spleen cells were seeded into plate with a density of $4 \times 10^5$ per well. Additionally, 10 μL OVA peptide solution (50 ng mL$^{-1}$) was added to each well. The culture plate was cultured overnight at 37 °C and 5% CO$_2$. After 24 h, the chromogenic reaction was performed according to the manufacturer's protocol.

## Inhibition of 4T1 tumor metastasis

We established 4T1 breast tumors to evaluate the therapeutic effects of aAGd-NWs in inhibiting 4T1 tumor metastasis. 4T1 breast tumor cells ($5 \times 10^5$ cells per mouse) were injected subcutaneously into the right lower abdomen to establish primary 4T1 tumors. When the tumor volume reached 80–100 mm$^3$, the mice were divided into five groups and were given Vehicle, Vehicle+RT, GGd-NCPs+RT, aAGd-NWs+RT, aAGd-NWs+RT + αPD-L1 ([Gd] = 15.7 mg kg$^{-1}$ and [ara-AMP] = 34.7 mg kg$^{-1}$ and [αPD-L1] = 10 mg kg$^{-1}$). As shown in the schematic illustration, the i.v. administration was given on day 0 and day 6, radiotherapy (5 Gy × 2) was given 6 h after injection, and anti-PD-L1 antibody (10.0 mg kg$^{-1}$ × 4 with fractions delivered 3 days apart, clone: 10 F.9G2, BioXcell, USA) was injected intraperitoneally 6 h after radiotherapy. The tumor volume and body weights were recorded daily during the 15-day treatment period. Then, the tumors tissues and surrounding epidermis were carefully removed and sutured. The

survival status and body weights of the mice were recorded for up to 200 days. Then lung tissues fixed with Bouin's solution were obtained for pathologic analysis when the mice died.

## Statistics and reproducibility

No statistical method was used to predetermine sample size. No data were excluded from the analyses. The experiments were randomized and the Investigators were not blinded to allocation during experiments and outcome assessment. Statistical analysis was performed by using two-tailed Student's $t$ test for two groups and one-way ANOVA analysis of variance for multiple groups. Then, $p$ values > 0.05 represented nonsignificance (N.S.) and $p$ values < 0.05 represented statistically significant, $*p < 0.05$, $**p < 0.01$, $***p < 0.001$. Error bars indicate the standard deviation (S.D.). Tumor size and weight of tumor-bearing mice were recorded with Microsoft Office 2019. GraphPad Prism Version 9.0.2 was used to analyze statistical data. FlowJo (Version 10.8.1) was used to analyze flow cytometry data. CaseViewer Version 2.4 and ImageJ Version 1.52 v were used to analyze immunofluorescent and immunochemical data. The pharmacokinetic parameters of ara-AMP were analyzed by DAS 2.1.1 software. Adobe Illustrator 2020 was used to create Figs. 1 and 2b. BioRender.com was used to create Figs. 2a, 6a, 8a and 9c. Calculate the minimum value of the conformational energy (MM2) of AMP and ara-AMP by Chem 3D 20.0.

## Reporting summary

Further information on research design is available in the Nature Portfolio Reporting Summary linked to this article.

## Data availability

The authors declare that data supporting the findings of this study are available within the article, and its Supplementary Information files. Source data are provided with this paper.

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

## Acknowledgements

We acknowledge the National Key Research and Development Program of China (No. 2022YFC34011603, A. Yuan), the National Natural Science Foundation of China (32371395, A. Yuan; 22275096, Z. Luo), Natural Science Foundation of Jiangsu Province (BK20221445, A. Yuan), Natural Science Key Fund for Universities in Jiangsu Province (22KJA430007, Z. Luo), General project of natural science research in colleges and universities of Jiangsu Province (22KJB350001, Z. Huang), the Central Fundamental Research Funds for the Central Universities, Natural Science Research Start up Foundation of Recruiting Talents of Nanjing University of Posts and Telecommunications (NY221133, Z. Huang), and the Project of State Key Laboratory of Organic Electronics and Information Displays, Nanjing University of Posts and Telecommunications (No. GZR2022010033, Z. Huang). Special thanks to Xiaoya Cui, Jiayue Su and Tao Liu for their contributions to cryo-EM structural analysis.

## Author contributions

Z. Huang, Z. Luo, Y. Hu, and A. Yuan, conceived the project, designed the studies and analyzed the results. Z. Huang, R. Gu, S. Huang, Q. Chen, J. Yan, H. Jiang, D. Yao, and C. Shen carried out the experiments. X. Cui, J. Su, and T. Liu, performed cryo-EM structural analysis. Z. Huang, R. Gu, and A. Yuan, wrote the manuscript. J. Wu, Z. Luo, Y. Hu, and A. Yuan, supervised and revised the project.

## Competing interests

The authors declare no competing interests.
