## [Peer Review File · Nature Communications]

REVIEWER COMMENTS

Reviewer #1: Gd-based nanomaterials:

This manuscript reports an interesting nanostructure with the ability to enhance radiation therapy and induce tumor activation. The authors performed thorough in vitro and in vivo assessments of the nanowires. The nanowires showed excellent efficacy for enhanced radiation therapy for cancer treatment. It is of interest to the readers in the field and can be published with minor revisions.

How stable are the chelation structure and the nanowire of aAGd-NWs?

What equipment was used for X-ray irradiation?

Figure 3: The coordination number of Gd(III) is 9. The structure showed in the figure only had 4 coordination binds. The proposed structure mat not reflect the true coordination structure in the nanowires.

Figure 5: The aAGd-NWs shows better penetration into the tumor 3d spheroids than GGd-NCPs. However, it only shows up to 70 μm . How about the penetration around the 200-400 μm into the 3D spheroids?

The dose of Gd in the aAGd-NWs for i.v. injection is 0.1 mmol/kg, similar to the clinical contrast agent for in vivo use. Given the potential low stability of the chelates, it should be a concern for clinical development, even the authors indicated acceptable safety in the discussions. It should be further discussed.

The grammatic errors should be corrected.

Reviewer #2: radiotherapy, immunotherapy and nanotechnology

In this study by Hu and colleagues, the authors synthesized chiral and ultrafine aAGd-NWs through a coordination-driven self-assembly method, using the clinically available ara-AMP and High-Z element Gd.

These nanowires demonstrated a remarkable ability to sensitize radiation therapy (RT), resulting in potent immunogenic cell death (ICD) and acting as an effective in-situ vaccine. Although the authors conducted numerous biological experiments, the material design lacks novelty, and the overall perspective of the manuscript is not sufficiently innovative.

(1) The language and presentation of the manuscript at times are difficult to understand. The clarity of the manuscript could use some improvement.

(2) What are the merits of using right-handed chirality aAGd-NWs in radiation therapy? Is it possible to modify the synthesis conditions to produce left-handed chirality aAGd-NWs? If achievable, a comparison between the two chiral forms of the material should be conducted in the context of tumor treatment.

(3) How can one confirm that only hydroxyl radicals are generated by aAGd-NWs during radiation therapy, and what is the principle behind the production of these radicals? The authors should conduct Electron Spin Resonance (ESR) tests to investigate the potential generation of other free radicals.

(4) What is the hydrated particle size of aAGd-NWs, and how stable are they in water, PBS, and culture medium environments? Do the particle sizes change over time, and is there any tendency for aggregation in their morphology?

(5) What is the principle behind the acid-responsive release of ara-AMP by aAGd-NWs? Since they can be released under acidic conditions, is the material degradable? Monitoring can be done through changes in particle size and TEM morphology.

(6) Flow cytometry experiments should be conducted to detect the apoptosis status of CT26 cells.

(7) The distribution of aAGd-NWs in major organs should be investigated, with a focus on identifying the main accumulation site.

(8) Hepatotoxicity should be monitored.

(9) The manuscript should thoroughly discuss and compare the results with the existing related literature to highlight its unique contributions. The significance of the research should be clearly outlined to establish its importance and relevance in the context of current scientific knowledge.

Reviewer #3: nanomedicine, immunotherapy

In this manuscript, Huang and colleagues reported a coordination strategy aAGd-NWs, comprising High-Z metal Gd and nucleoside prodrug ara-AMP combining radiotherapy as an in situ tumor vaccine. The aAGd-NWs exhibited excellent anti-tumor effect and stimulated anti-tumor immunity. However, there are some issues in the investigation of anti-tumor mechanism. Some viewpoints lack solid proof to support them. The manuscript can only be published after solving the following issues.

1. Please provide the stability of aAGd-NWs under different conditions.
2. The biodistribution of aAGd-NWs in different organs should be studied.
3. Fig. 7c, please also provide the expression level of total IRF3 and STING. And it seems that ara-AMP largely contributed to the STING pathway and ICD induction, but the ara-AMP function was not well examined. I suggest to add single ara-AMP group or ara-AMP+RT group to explain the combinative effect better.
4. The in vitro and in vivo immunostimulation of APCs by aAGd-NWs should be sufficiently verified.
5. Fig. 8e, as shown in the illustration, DCs crosstalk with naive T cells and promote T cell activation, proliferation, and migration, but only T cell activation and migration were investigated. Similarly, the illustration shows that T effector cells secrete perforin, granzyme B and IFN- γ to kill cancer cells, but only IFN- γ secretion was investigated. Please delete the irrelative contents, which may confuse the readers.
6. The author examined the aAGd-NWs-sensitized radiation-induced systemic immune response in CT26 tumor model only by analyzing matured DCs in TDLNs and infiltrating T cells in tumor sites. More solid evidences are needed to demonstrate the systemic anti-tumor immunity after injection of aAGd-NWs and local radiation of tumor area.
7. The gating strategy of flow cytometric analyses should be provided.

Reviewer #4: cancer immunology, ICD

Raw present manuscript by Hu and colleagues describes the generation of chiral vidarabine monophosphate-gadolinium nanowires (aAGd-NWs) that enhance X-ray deposition, inhibits DNA repair and induces the release of immunogenic cell death (ICD)-related danger associated molecular patterns (DAMPs). In vivo aAGd-NWs combined with radiotherapy induce T cell-dependent anticancer immunity that sensitises to immune checkpoint inhibition (ICI).

The present work extends on a recent paper (PMID 33420008) by the same group employing an identical strategy and methodology.

Similar to the work published before mechanistic insights are missing and the work stays largely on a descriptive level.

General comments: Representative images of negative controls (with and w/o irradiation; with and w/o aAGd-NW) are missing almost entirely throughout the paper. The figure legends need to describe important experimental settings such as the method used to evaluate a certain parameter (such as viability), the time point at which samples have been collected and the way data has been normalised.

Linguistic editing would help to improve readability.

Specific comments:

Colocalization in Fig.4a needs to be quantified. In fact the images indicate at best a partial localisation of the NWs to lysosomes.

How was viability assessed in Fig. 4f and g and why does radiotherapy (RT) alone cause so little toxicity?

To my impression the ring-like aggregation of fluorescence all around the structure in Fig 5a seems to indicate the opposite of what the authors claim namely the inability of the NWs to penetrate the spheroids. Some clarification or positive controls would be useful.

Fig.5 d-f again indicate only very little toxicity by RT alone.

Fig.6 h,i depict values in percent of what? It might be advisable to use another means of quantification.

The band pattern in Fig7 pIRF3 and pSTING is different (despite exact same molecular weight) indicating that only one loading control is not appropriate. In any case phosphonoepitope-specific antibodies should have non-phosphonoepitope-specific antibodies (used on the same stripped membrane) of the same protein as control.

Further suggestions:

RT has been described to emit DAMPs which is not the case in this work. This should be discussed.

aAGd-NWs should be benchmarked against other radiosensitizers and ICD inducers.

Response to Referees

Dear Reviewers:

Thank you for your constructive comments concerning our manuscript entitled “Chiral Coordination Polymer Nanowires Boost Radiation-Induced *In Situ* Tumor Vaccination” (ID: NCOMMS-23-28217). These comments are all invaluable and have been instrumental in the revision and enhancement of our paper, providing significant guidance for our research. We have examined the comments carefully and made corrections that we hope will meet with your approval. Revisions are highlighted in yellow in the paper. The point to point responses to the reviewer's comments are as following:

Reviewer #1 (Gd-based nanomaterials):

Comment 1: This manuscript reports an interesting nanostructure with the ability to enhance radiation therapy and induce tumor activation. The authors performed thorough *in vitro* and *in vivo* assessments of the nanowires. The nanowires showed excellent efficacy for enhanced radiation therapy for cancer treatment. It is of interest to the readers in the field and can be published with minor revisions.

Response: Thank you very much for your insightful comments. We have carefully revised the manuscript according to your comments and suggestions. All the changes are highlighted in yellow. The Point-by-point responses to your comments are listed below.

Comment 2: How stable are the chelation structure and the nanowire of aAGd-NWs?

Response: We appreciate the reviewer for the comment. We carefully studied the stability of the chelated structure of aAGd-NWs under room temperature (PBS, 25 °C) and simulated physiological conditions (10% Fetal Bovine Serum, 37 °C). Transmission electron microscopy (TEM) imaging demonstrates that the width of aAGd-NWs measures approximately 1.3 nm and the length varies from 50 to 200 nm. Due to Brownian motion, the dynamic light scattering (DLS) method can only determine the lengths of the aAGd-NWs, and the particle size distribution is primarily

between 150 and 200 nm.

Fig. R1 Stability of aAGd-NWs. (a) Dynamic light scattering (DLS) data of aAGd-NWs diluted with PBS at 25 °C or 10% fetal bovine serum at 37 °C, respectively (n = 3). (b) Representative UV-Vis spectra of aAGd-NWs in PBS (25 °C) at different time points (n = 3). (c) Schematic illustration of ara-AMP secondary acid dissociation. (d) Proposed mechanism of the pH dependent degradation process of aAGd-NWs.

As depicted in **Fig. R1a**, aAGd-NWs maintain a relatively stable particle size distribution (< 250 nm) when diluted by H₂O (25 °C) and 10% Serum (37 °C), signifying that aAGd-NWs exhibit robust stability under both room temperature storage and simulated physiological conditions. Additionally, the UV absorption spectrum at 72 hours correlates with the initial UV spectrum, further corroborating the exceptional stability of aAGd-NWs (**Figs. R1b, c**). Considering that aAGd-NWs release drugs in a pH-dependent manner, we hypothesize that the coordination state of Gd in aAGd-NWs was closely linked to the pK_a of ara-AMP (pK_{a1} = 3.8, pK_{a2} = 6.2) (**Fig. R1d**)¹. Theoretically, when pH > 6.2, aAGd-NWs maintain a relatively stable

particulate state. As the pH value gradually decreased, the phosphate partially became mono-protonated ($3.8 < \text{pH} \leq 6.2$), these nanowires would still maintain their particulate or coordination state. When $\text{pH} \leq 3.8$, free Gd^{3+} is hypothesized to be progressively released from aAGd-NWs owing to the further protonation of phosphate groups (**Fig. R1e**). These findings are presented in Line 155-161 and 663-677 of the Manuscript, and are further illustrated in Supplementary Figure 1i, j and Supplementary Figure 18a, b of the Supplementary Information.

1. Kwee, M. S. L., Stolk, L. M. L. Formulation of a stable vidarabine phosphate injection. *Pharmaceutisch Weekblad* **6**, 101-104 (1984).

Comment 3: What equipment was used for X-ray irradiation?

Response: Thank you for the question. The X-ray irradiation equipment utilized in our study is the Rad Source RS2000 X-ray irradiator (Model: X-ray Irradiator RS2000, USA). The instrument's technical specifications comprise a dose rate of 10 Gy min^{-1} for cells and 1.2 Gy min^{-1} for small animals. The irradiation plane is situated 25 cm away in the cone-shaped irradiation field. The operating specifications of the power supply are 220 volts, 60 Hz, 40 amps. In response to the reviewer's comments, we have incorporated a detailed description of the irradiation equipment, highlighted within the Experimental Methods section (Line 769-774).

Comment 4: Figure 3: The coordination number of Gd (III) is 9. The structure showed in the figure only had 4 coordination binds. The proposed structure mat not reflect the true coordination structure in the nanowires.

Response: We must express our gratitude to the reviewer for the insightful comments. As correctly pointed out by the reviewer, the coordination number of Gd (III) ranges from 8 to 9. Based on simulation results, we delineate the plausible assembly mechanism for aAGd-NWs in three distinct steps: in the first step, ara-AMP and Gd initially self-assemble into supramolecular macrocyclic structures, which sequentially stack *via* coordination-driven self-assembly, creating four coordination bonds. In the second step, the assembling macrocyclic structure undergoes a 180° rotation and

coordinates with the existing structures, thereby creating additional two coordination bonds and resulting in the observed right-handed chirality of aAGd-NWs. Then, the addition of 2% MSA (dissolved in 0.9% NaCl), serving as emulsifier and aggregation inhibitor, lead to uniformly dispersed and stable aAGd-NWs. Consequently, it is reasonable to postulate that Cl⁻ is also involved in the coordination of Gd (**Fig. R2**). Following the reviewer's constructive suggestions, we have revised the proposed structural schematics and elaborated on this content in the manuscript (**Fig. 3 and Line 216-219**) to more accurately represent the true coordination structure of the nanowires.

Fig. R2 Structural analysis of aAGd-NWs. (a) Circular dichroism (CD) spectrum of aAGd-NWs. (b) Calculated images of aAGd-NWs based on 2D imaging. (c) Calculated 3D structure based on 3D modeling reconstruction. (d, e) Proposed self-assembly mechanism of aAGd-NWs.

Comment 5: Figure 5, The aAGd-NWs shows better penetration into the tumor 3D spheroids than GGd-NCPs. However, it only shows up to 70 μm . How about the penetration around the 200-400 μm into the 3D spheroids?

Response: Thanks for the reviewer's constructive comments. Due to the light absorption and scattering properties inherent in most biological samples, the penetration depth of confocal laser microscopy is typically limited to approximately

100 μm^2 . To delve deeper into the penetration capabilities of ultrafine aAGd-NWs in 3D CT26 spheroids, we dissociated these spheroids into single cells for flow cytometry analysis. As illustrated in **Fig. R3**, compared to Vehicle-treated 3D cell spheroids (2.2%), the uptake rate of ICG@aAGd-NWs in tumor cells within 3D spheroids was significantly higher at 79.7%. These findings suggest that ultrafine aAGd-NWs possess the capacity to deeply penetrate the spheroids and effectively infiltrate a substantial proportion of tumor cells. These findings are presented in Line 293-302 of the Manuscript, and are further illustrated in Supplementary Figure 5 of the Supplementary Information.

Fig. R3 Flow cytometry analysis of the dissociated tumor cells from spheroids treated by Vehicle or ICG@aAGd-NWs for 24 hours.

2. Jonkman, J., Brown, C. M., Wright, G. D. et al. Tutorial: guidance for quantitative confocal microscopy. *Nat. Protoc.* **15**, 1585-1611 (2020).

Comment 6: The grammatic errors should be corrected.

Response: Thank you for bringing this to our attention. We appreciate your feedback regarding the grammatical errors in the manuscript. We have carefully reviewed and corrected any grammatical errors to ensure the quality and readability of the manuscript with the help of *Nature Research Editing Service*.

Special thanks to Reviewer #1 for his/her comments. These comments have significantly improved the quality of this paper.

Reviewer #2 (radiotherapy, immunotherapy and nanotechnologies):

Comment 1: In this study by Hu and colleagues, the authors synthesized chiral and ultrafine aAGd-NWs through a coordination-driven self-assembly method, using the clinically available ara-AMP and High-Z element Gd. These nanowires demonstrated a remarkable ability to sensitize radiation therapy (RT), resulting in potent immunogenic cell death (ICD) and acting as an effective in-situ vaccine. Although the authors conducted numerous biological experiments, the material design lacks novelty, and the overall perspective of the manuscript is not sufficiently innovative.

Response: Thanks for the constructive comments. We have accordingly enhanced the discussion on the innovative aspects of this study. The primary objective of this study is to harness the self-assembly of clinical drugs for self-delivery purposes and to facilitate radiation-induced *in situ* tumor vaccination. We aim to elucidate two primary innovations. **From the materials perspective:** Firstly, we successfully synthesized ultrafine chiral nanowires (aAGd-NWs) through the self-assembly of clinical drug molecules, independent of any template usage. Secondly, we employed cryogenic transmission electron microscopy to analyze the structure of the ultrafine nanowires and unravel the drug molecules' self-assembly mechanism *via* theoretical

simulation. **From the biological perspective:** The synthetic ultrafine nanowires impart them with substantial biological significance. Firstly, the ultrafine structure of the nanowires we engineered facilitates deep penetration of drugs into tumor tissues, thereby augmenting therapeutic efficacy. Secondly, high-Z ultrafine nanowires not only exploit their high atomic number element for enhanced X-ray deposition and scattering, thus increasing radiotherapy sensitivity, but also release ara-AMP in tumor tissues, inhibiting tumor cell DNA repair and promoting damaged DNA release to activate the cGAS-STING pathway, inducing a potent *in situ* tumor vaccination. Furthermore, the development of self-delivery nanowires through the self-assembly of clinical drugs holds the potential to accelerate their clinical translation by obviating the need for nanocarriers. These discussions are presented in Line 625-636 of the Manuscript.

Comment 2: The language and presentation of the manuscript at times are difficult to understand. The clarity of the manuscript could use some improvement.

Response: Thank you for bringing this to our attention. We appreciate your feedback regarding the language and presentation of the manuscript. We carefully polished the English expression again with the help of *Nature Research Editing Service*.

Comment 3: What are the merits of using right-handed chirality aAGd-NWs in radiation therapy? Is it possible to modify the synthesis conditions to produce left-handed chirality aAGd-NWs? If achievable, a comparison between the two chiral forms of the material should be conducted in the context of tumor treatment.

Response: Thanks for the reviewer's constructive questions and suggestions. We employed various methods and conducted experiments to pursue the synthesis of left-handed chirality aAGd-NWs. Despite extensive efforts, this goal has not yet been achieved. As demonstrated in **Fig. R4a**, modifications the self-assembly ratio of ara-AMP and Gd, and adjustments in the reaction temperature, appear to have no impact on the circular dichroism (CD) values of ara-AMP-Gd NCPs.

Fig. R4 Synthesis and CD characterization. (a) Synthesis and CD characterization of aAGd-NWs. (b) Synthesis and CD characterization of AGd-NCPs.

Conversely, substituting the ligand molecule from ara-AMP to AMP (Adenosine monophosphate) led to a notable reversal in the CD values, irrespective of the self-assembly ratios and reaction temperatures. These findings indicate that the chirality of nanoscale coordination polymers is predominantly influenced by the ligand molecules, rather than changes in external conditions. Then, AMP and Gd^{3+} self-assemble to form spherical nanoscale coordination polymers (AGd-NCPs). This phenomenon is attributed to the fact that in AMP coordination, the C-N bond connecting the adenine group and the ribose group in the AMP molecule is not constrained by rotation, resulting in anisotropic amorphous structures that eventually assemble into spherical particles. In contrast, the steric hindrance of the 3'-OH group in the ara-AMP molecule restricts the free rotation of the C-N bond between the

adenine and ribose groups, maintaining a rigid conformation that leads to cooperative stacking and the formation of nanowires. These findings are presented in Line 697-707 of the Manuscript, and are further illustrated in Supplementary Figure 22 of the Supplementary Information.

Comment 4: How can one confirm that only hydroxyl radicals are generated by aAGd-NWs during radiation therapy, and what is the principle behind the production of these radicals? The authors should conduct Electron Spin Resonance (ESR) tests to investigate the potential generation of other free radicals.

Response: Thanks for the reviewer's constructive comments. X-ray irradiation primarily acts on water molecules and produce hydroxyl radicals ($\bullet\text{OH}$) to cause cellular damage. In our study, we used MB decay to detect the Reactive Oxygen Species (ROS) generated by RT or aAGd-NWs sensitized RT. Previous studies have confirmed that MB decomposition rate correlates with $\bullet\text{OH}$ formation^{3,4}. Upon exposure to radiation, high-Z elements absorb energy and undergo multiple processes, releasing particles, such as Compton electrons and photoelectrons, which react with organic molecules or water, leading to their radioactive decomposition and the formation of $\bullet\text{OH}$ ⁵.

Fig. R5 Electron spin resonance (ESR) spectra of singlet oxygen detection for different treatments. 2,2,6,6-Tetramethyl-4-piperidinone (TEMP) was used as a trap for singlet oxygen ($^1\text{O}_2$).

Subsequently, following the reviewer's insightful recommendation, we investigated the formation of other ROS using electron spin resonance (ESR) spectroscopy. As illustrated in the **Fig. R5**, TEMP was used as a trap for singlet oxygen ($^1\text{O}_2$) to produce TEMPO, which was quantified by ESR spectroscopy. The results obtained are in alignment with previously reported findings⁶. Radiotherapy (RT) can induce the production of a measurable amount of singlet oxygen, yet aAGd-NWs exert no significant influence on this process. These findings are presented in Line 183-189 of the Manuscript, and are further illustrated in Supplementary Figure 1k of the Supplementary Information.

3. Yang, B., Chen, Y., Shi, J. Reactive oxygen species (ROS)-based nanomedicine. *Chem. Rev.* **119**, 4881-4985 (2019).

4. Dou, Y., Liu, Y., Zhao, F. et al. Radiation-responsive scintillating nanotheranostics for reduced hypoxic radioresistance under ROS/NO-mediated tumor microenvironment regulation. *Theranostics* **8**, 5870-5889 (2018).

5. Wang, H., Mu, X., He, H., Zhang, X.-D. Cancer radiosensitizers. *Trends Pharmacol. Sci.* **39**, 24-48 (2018).

6. Chang, Y., Huang, J., Shi, S. et al. Precise engineering of a Se/Te nanochaperone for reinvigorating cancer radio-immunotherapy. *Adv. Mater.* **35**, 2212178 (2023).

Comment 5: What is the hydrated particle size of aAGd-NWs, and how stable are they in water, PBS, and culture medium environments? Do the particle sizes change over time, and is there any tendency for aggregation in their morphology?

Response: Thanks for your valuable comments. As depicted in **Fig. R6a, b**, the hydrated particle size of aAGd-NWs is 189 nm, aligning with the length (150~200 nm) observed through TEM, and demonstrates relatively high stability in water (25 °C), PBS (25 °C) and 10% FBS (37 °C). Additionally, the UV absorption spectrum at 72 hr aligned with the initial UV spectrum (**Fig. R6c**), further confirming the stability of aAGd-NWs. Notably, aAGd-NWs demonstrated no significant aggregation trend over time. These findings are presented in Line 155-161 of the Manuscript, and are further illustrated in Supplementary Figure 1h-j of the Supplementary Information.

Fig. R6 Stability of aAGd-NWs. (a) Hydrated particle size of aAGd-NWs (b) DLS data of aAGd-NWs diluted with water (25 °C), PBS (25 °C) or 10% FBS (37 °C), respectively (n = 3). (c) Representative UV-vis spectra of aAGd-NWs in PBS (25 °C) at different time points.

Comment 6: What is the principle behind the acid-responsive release of ara-AMP by aAGd-NWs? Since they can be released under acidic conditions, is the material degradable? Monitoring can be done through changes in particle size and TEM morphology.

Response: Thanks for your valuable suggestions. Considering that aAGd-NWs release drugs in a pH-dependent manner, we speculated that the coordination state of Gd in aAGd-NWs was closely linked to the pK_a of ara-AMP (pK_{a1} = 3.8, pK_{a2} = 6.2) (Fig. R7a)¹. Theoretically, when pH > 6.2, aAGd-NWs maintained a relatively stable particulate state. As the pH value gradually decreased, the phosphate partially became mono-protonated (3.8 < pH ≤ 6.2), these nanowires would still maintain their particulate or coordination state. When pH ≤ 3.8, free Gd³⁺ may be progressively released from aAGd-NWs owing to the further protonation of phosphate groups (Fig. R7b). As shown in the Fig. R7c, aAGd-NWs exhibited robust stability under physiological condition (pH = 7.4). However, with the pH value decreasing (pH = 5.0), aAGd-NWs will gradually degrade, which was confirmed by high-resolution transmission electron microscopy (HRTEM) (Fig. R7d). These findings are presented in Line 663-677 of the Manuscript, and are further illustrated in Supplementary Figure 18 of the Supplementary Information.

Fig. R7 Degradation mechanism of aAGd-NWs. (a) Schematic illustration of ara-AMP secondary acid dissociation. (b) Proposed mechanism of the pH dependent degradation process of aAGd-NWs. (c) DLS data of aAGd-NWs in HEPES buffer containing 10% serum at different pH values (7.4, 5.0) *in vitro* (n = 3). (d) High-resolution transmission electron microscopy (HRTEM) images of aAGd-NWs incubated for 48 hours under different pH values (7.4, 5.0) *in vitro*.

1. Kwee, M. S. L., Stolk, L. M. L. Formulation of a stable vidarabine phosphate injection. *Pharmaceutisch Weekblad* **6**, 101-104 (1984).

Comment 7: Flow cytometry experiments should be conducted to detect the apoptosis status of CT26 cells.

Response: Thanks for the reviewer's constructive comments. According to the reviewer's insightful suggestion, we performed flow cytometry to assess the apoptosis status of treated CT26 cells. As illustrated in the **Fig. R8**, the flow cytometry results of PI and Annexin V staining indicate that aAGd-NWs-sensitized RT promotes more apoptosis, suggesting that the aAGd-NWs significantly enhance the efficacy of

radiotherapy in killing tumor cells. These findings are presented in Line 277-280 of the Manuscript, and are further illustrated in Supplementary Figure 4 of the Supplementary Information.

Fig. R8 Flow cytometry analysis of apoptosis status of treated CT26 cells.

Comment 8: The distribution of aAGd-NWs in major organs should be investigated, with a focus on identifying the main accumulation site.

Response: Thanks for the reviewer's constructive comments. Building on our previous work examining the pharmacokinetics of aAGd-NWs, we conducted a detailed analysis of their distribution in major organs using ICP-OES. As depicted in the **Fig. R9**, 6 hours post aAGd-NWs injection, the highest concentration was observed in tumor tissues ($23.6 \mu\text{g g}^{-1}$), followed by distribution in renal tissues ($12.3 \mu\text{g g}^{-1}$), with smaller quantities detected in the heart, liver, spleen, and lungs. Importantly, the presence of free Gd ions could not be directly detected within tissues that have not been incinerated and nitrified. This finding indicates that post metabolic processes *in vivo*, Gd remains either coordinated or partially coordinated state rather than in a free state, which implies a preliminary indication of biological safety. These

findings are presented in Line 347-355 of the Manuscript, and are further illustrated in Supplementary Figure 6 of the Supplementary Information.

Fig. R9 The biodistribution of aAGd-NWs in major organs detected by ICP-OES (n = 3).

Comment 9: Hepatotoxicity should be monitored.

Response: In response to the reviewers' constructive comments, we performed further serum biochemical analyses to assess liver and kidney toxicity.

Fig. R10 Serum biochemical analysis. (a) Alanine aminotransferase (ALT), (b) aspartate aminotransferase (AST), (c) creatinine (CREA) and (d) urea (UREA) of CT26-bearing mice after various treatments (n = 3). All data were shown as mean ± SD. Statistical significance

was determined using two-tailed Student's *t*-test for pairwise comparisons, and one-way ANOVA analysis of variance for multiple groups. *p* values > 0.05 were considered non-significant (N.S.), while *p* values < 0.05 were considered statistically significant.

Independent of radiotherapy exposure, free ara-AMP is known to induce liver and kidney toxicity. However, upon self-assembly with Gd, it exhibits selective release and activity in tumor tissues, markedly reducing its hepatotoxic and nephrotoxic effects (**Fig. R10**). These findings are presented in Line 689-696 of the Manuscript, and are further illustrated in Supplementary Figure 20 of the Supplementary Information.

Comment 10: The manuscript should thoroughly discuss and compare the results with the existing related literature to highlight its unique contributions. The significance of the research should be clearly outlined to establish its importance and relevance in the context of current scientific knowledge.

Response: Thanks for the constructive comments. We have accordingly enhanced the discussion on the innovative aspects of this study. The primary objective of this study is to harness the self-assembly of clinical drugs for self-delivery purposes and to facilitate radiation-induced in situ tumor vaccination. We aim to elucidate two primary innovations. **From the materials perspective:** Firstly, we successfully synthesized ultrafine chiral nanowire drugs (aAGd-NWs) through the self-assembly of clinical drug molecules, independent of any template usage. Secondly, we employed cryogenic transmission electron microscopy to analyze the structure of the ultrafine nanowires and unravel the drug molecules' self-assembly mechanism via theoretical simulation. **From the biological perspective:** The synthetic ultrafine nanowires impart them with substantial biological significance. Firstly, the ultrafine structure of the nanowires we engineered facilitates deep penetration of drugs into tumor tissues, thereby augmenting therapeutic efficacy. Secondly, high-Z ultrafine nanowires not only exploit their high atomic number element for enhanced X-ray deposition and scattering, thus increasing radiotherapy sensitivity, but also release

ara-AMP in tumor tissues, inhibiting tumor cell DNA repair and promoting damaged DNA release to activate the cGAS-STING pathway, inducing a potent *in situ* tumor vaccination. Finally, the development of self-delivery nanowires through the self-assembly of clinical drugs holds the potential to accelerate their clinical translation by obviating the need for nanocarriers. These discussions are presented in Line 625-636 of the Manuscript.

Special thanks to Reviewer #2 for his/her good comments. These comments have significantly improved the quality of this paper.

Reviewer #3 (nanomedicine, immunotherapy):

Comment 1: In this manuscript, Huang and colleagues reported a coordination strategy aAGd-NWs, comprising High-Z metal Gd and nucleoside prodrug ara-AMP combining radiotherapy as an *in situ* tumor vaccine. The aAGd-NWs exhibited excellent anti-tumor effect and stimulated anti-tumor immunity. However, there are some issues in the investigation of anti-tumor mechanism. Some viewpoints lack solid proof to support them. The manuscript can only be published after solving the following issues.

Response: Thank you very much for your insightful comments. We have carefully revised the manuscript according to your comments and suggestions. All the changes are highlighted in yellow. Point-by-point responses are listed below.

Comment 2: Please provide the stability of aAGd-NWs under different conditions.

Response: Thanks for your valuable comments. As depicted in **Figs. R11a, b**, the average hydrated particle size of aAGd-NWs is 189 nm, aligning with the length (150~200 nm) observed through TEM, and demonstrates relatively high stability in water (25 °C), PBS (25 °C) and 10% FBS (37 °C). Additionally, the UV absorption spectrum at 72 hr was consistent with the initial UV spectrum (**Fig. R11c**), further confirming the stability of aAGd-NWs. Notably, aAGd-NWs demonstrated no significant aggregation trend over time. These findings are presented in Line 155-161 of the Manuscript, and are further illustrated in Supplementary Figure 1h-j of the

Supplementary Information.

Fig. R11 Stability of aAGd-NWs. (a) Hydrated particle size of aAGd-NWs (b) DLS data of aAGd-NWs diluted with water (25 °C), PBS (25 °C) or 10% FBS (37 °C), respectively (n = 3). (c) Representative UV-vis spectra of aAGd-NWs in PBS (25 °C) at different time points.

Comment 3: The biodistribution of aAGd-NWs in different organs should be studied.

Response: Thanks for the reviewer's constructive comments. Building on our previous work examining the pharmacokinetics of aAGd-NWs, we conducted a detailed analysis of their distribution in major organs using ICP-OES. As depicted in the **Fig. R12**, 6 hours post aAGd-NWs injection, the highest concentration was observed in tumor tissues ($23.6 \mu\text{g g}^{-1}$), followed by distribution in renal tissues ($12.3 \mu\text{g g}^{-1}$), with smaller quantities detected in the heart, liver, spleen, and lungs. Importantly, the presence of free Gd ions could not be directly detected within tissues that have not been incinerated and nitrified. This finding indicates that post metabolic processes *in vivo*, Gd remains either coordinated or partially coordinated state rather than in a free form, thereby implying initial biological safety. These findings are presented in Line 347-355 of the Manuscript, and are further illustrated in Supplementary Figure 6 of the Supplementary Information.

Fig. R12 The biodistribution of aAGd-NWs in major organs detected by ICP-OES.

Comment 4: Fig. 7c, please also provide the expression level of total IRF3 and STING. And it seems that ara-AMP largely contributed to the STING pathway and ICD induction, but the ara-AMP function was not well examined. I suggest to add single ara-AMP group or ara-AMP+RT group to explain the combinative effect better.

Response: We highly appreciate the reviewer's valuable suggestion. The expression levels of total IRF3 and STING are now presented in Fig. 7c of the manuscript (Fig. R13a). As illustrated in **Fig. R13b**, co-culture of ara-AMP or RT treated CT26 tumor cells with immune cells led to relatively weak STING and interferon regulatory factor 3 (IRF-3) phosphorylation. This observation indicates that ara-AMP itself possesses a relatively weak capability to activate the cGAS/STING pathway, comparable to radiation therapy. These findings are presented in Line 422-426 and Fig. 7d of the Manuscript, and are further illustrated in Supplementary Figure 8 of the Supplementary Information.

Fig. R13 Activation of cGAS-STING. (a) Western blot of Vehicle, GGd-NCPs, aAGd-NWs,

Vehicle+RT, GGd-NCPs+RT, aAGd-NWs+RT groups. (b) Western blot of Vehicle, ara-AMP, RT and ara-AMP+RT groups.

Comment 5: The *in vitro* and *in vivo* immunostimulation of APCs by aAGd-NWs should be sufficiently verified.

Response: Thank you for your constructive comments. We recognize the criticality of meticulously validating the immunostimulation of APCs by aAGd-NWs sensitized radiation *in vivo*. We further conducted additional experiments to strengthen the evidence supporting the immunostimulatory effects of aAGd-NWs+RT upon APCs. In *in vivo* experiments of OVA-specific antigen stimulation, the proportion of CD11c⁺ H2Kb-SIINFEKL⁺ dendritic cells (DCs) following aAGd-NWs+RT treatment was significantly higher than that in Vehicle, ara-AMP, GGd-NCPs with or without RT groups (**Fig. R14**). This finding substantiates that aAGd-NWs sensitize RT and synergistically facilitate *in situ* vaccine priming, thereby potentially augmenting systemic anti-tumor immune responses. These findings are presented in Line 501-506 and Figure 8f of the Manuscript, and are further illustrated in Supplementary Figure 11 of the Supplementary Information.

Fig. R14 The *in vivo* immunostimulation of APCs by aAGd-NWs sensitized RT. (a, b) Flow cytometry imaging and analysis of H2Kb-SIINFEKL⁺ CD11c⁺ DCs in tumor draining lymph nodes after Vehicle, ara-AMP, GGd-NCPs, aAGd-NWs, Vehicle+RT, ara-AMP+RT,

GGd-NCPs+RT, aAGd-NWs+RT treatments (n = 3). All data were shown as mean ± SD. Statistical significance was determined using two-tailed Student's *t*-test for pairwise comparisons, and one-way ANOVA analysis of variance for multiple groups. *p* values > 0.05 were considered non-significant (N.S.), while *p* values < 0.05 were considered statistically significant.

Comment 6: Fig. 8e, as shown in the illustration, DCs crosstalk with naive T cells and promote T cell activation, proliferation, and migration, but only T cell activation and migration were investigated. Similarly, the illustration shows that T effector cells secrete perforin, granzyme B and IFN- γ to kill cancer cells, but only IFN- γ secretion was investigated. Please delete the irrelative contents, which may confuse the readers.

Response: Thank you for your valuable suggestions. We have revised the manuscript to focus on the studied aspects of DCs and T cell interactions, particularly highlighting the secretion of IFN- γ by T lymphocytes for tumor cell eradication. In response to your concern, extraneous material has been excised from Fig. 8e to augment clarity and avert potential misinterpretation among readers (**Fig. R14**).

Fig. R14 Schematic representation of immune priming process and checkpoint blockade immunotherapy potentiation.

Comment 7: The author examined the aAGd-NWs-sensitized radiation-induced systemic immune response in CT26 tumor model only by analyzing matured DCs in TDLNs and infiltrating T cells in tumor sites. More solid evidences are needed to demonstrate the systemic anti-tumor immunity after injection of aAGd-NWs and local radiation of tumor area.

Response: Thank you for your insightful comment. In response to your invaluable

suggestion, we utilized the IFN- γ ELISpot (enzyme-linked immune absorbent spot) assay to detect the capacity of aAGd-NWs+RT in inducing systemic anti-tumor immunity. This assay facilitates the measurement of antigen-specific T cells in an immune sample, given that T lymphocytes secrete cytokines, such as interferon (IFN)- γ , upon binding their cognate antigen epitope. Immune cells are extracted from the spleens of aAGd-NWs+RT immunized mice and incubated with OVA₂₅₇₋₂₆₄ peptide (SIINFEKL) on polyvinylidene difluoride (PVDF)-lined microplates precoated with a capture antibody to IFN- γ , and cytokine spots are identified using an IFN- γ -specific detection antibody and an enzyme-linked conjugate. As depicted in the **Fig. R15a and b**, the number of OVA₂₅₇₋₂₆₄ peptide-specific CD8⁺ T cells in the splenocytes of mice treated by aAGd-NW+RT was significantly higher than those in the RT alone or GGd-NCPs+RT groups. These findings further substantiate that the combination of aAGd-NW and RT is exceptionally effective in inducing *in situ* vaccination and initiating antigen-specific anti-tumor immune responses. **These findings are presented in Line 546-557 and Fig. 9a, b of the Manuscript.**

Fig. R15 Antigen-specific T cells by IFN- γ ELISpot assay. (a) The representative images of antigen-specific T Cells by IFN- γ ELISpot. (b) The Quantification of antigen-specific T Cells by IFN- γ ELISpot (n = 3). All data were shown as mean \pm SD. Statistical significance was determined using two-tailed Student's *t*-test for pairwise comparisons, and one-way ANOVA analysis of variance for multiple groups. Then, *p* value > 0.05 were considered non-significant (N.S.), while *p* value < 0.05 were considered statistically significant.

Comment 8: The gating strategy of flow cytometric analyses should be provided.

Response: Thanks a lot for your reminder. In this study, all of the flow cytometry experiments were adopted with similar gating strategy. Forward Scatter (FSC) and Side Scatter (SSC) dot maps were established during the running process, and the

voltage was adjusted to ensure that all events were visible on the dot maps. Gating was then performed to select events with appropriate FSC (200-600) and SSC (200-600) values, while events with low FSC/low SSC and low FSC/high SSC were excluded as they represented cell debris and air bubbles. Specific cell types were gated using fluorescently labeled antibodies, including dendritic cells (CD80⁺ and CD86⁺ gated on CD11c⁺), CD4⁺ T cells (CD3⁺ and CD4⁺), CD8⁺ T cells (CD3⁺ and CD8⁺), OVA-specific dendritic cells (CD11c⁺ H2Kb-SIINFEKL⁺). These gating strategies have been integrated into Method Section (Line 985-994 of the Manuscript).

Special thanks to Reviewer #3 for his/her good comments. These comments have significantly improved the quality of this paper.

Reviewer #4 (cancer immunology, ICD):

Comment 1: Raw present manuscript by Hu and colleagues describes the generation of chiral vidarabine monophosphate-gadolinium nanowires (aAGd-NWs) that enhance X-ray deposition, inhibits DNA repair and induces the release of immunogenic cell death (ICD)-related danger associated molecular patterns (DAMPs). In vivo aAGd-NWs combined with radiotherapy induce T cell-dependent anticancer immunity that sensitises to immune checkpoint inhibition (ICI).

The present work extends on a recent paper (PMID 33420008) by the same group employing an identical strategy and methodology.

Similar to the work published before mechanistic insights are missing and the work stays largely on a descriptive level.

Response: Thanks for the constructive comments. We have accordingly enhanced the discussion on the innovative aspects of this study. The primary objective of this study is to harness the self-assembly of clinical drugs for self-delivery purposes and to facilitate radiation-induced in situ tumor vaccination. We aim to elucidate two primary innovations. **From the materials perspective:** Firstly, we successfully synthesized ultrafine chiral nanowire drugs (aAGd-NWs) through the self-assembly of clinical drug molecules, independent of any template usage. Secondly, we employed

cryogenic transmission electron microscopy to analyze the structure of the ultrafine nanowires and unravel the drug molecules' self-assembly mechanism via theoretical simulation. **From the biological perspective:** The synthetic ultrafine nanowires impart them with substantial biological significance. Firstly, the ultrafine structure of the nanowires we engineered facilitates deep penetration of drugs into tumor tissues, thereby augmenting therapeutic efficacy. Secondly, high-Z ultrafine nanowires not only exploit their high atomic number element for enhanced X-ray deposition and scattering, thus increasing radiotherapy sensitivity, but also release ara-AMP in tumor tissues, inhibiting tumor cell DNA repair and promoting damaged DNA release to activate the cGAS-STING pathway, inducing a potent *in situ* tumor vaccination. Finally, the development of self-delivery nanowires through the self-assembly of clinical drugs holds the potential to accelerate their clinical translation by obviating the need for nanocarriers. These discussions are presented in Line 625-636 of the Manuscript.

Comment 2: General comments: Representative images of negative controls (with and w/o irradiation; with and w/o aAGd-NW) are missing almost entirely throughout the paper. The figure legends need to describe important experimental settings such as the method used to evaluate a certain parameter (such as viability), the time point at which samples have been collected and the way data has been normalised.

Response: Thank you for your valuable feedback. Previously, we placed some control group data in the supplementary materials, leading to some misunderstandings. We addressed these issues by including representative images of negative controls in the main text. Additionally, we have modified the figure legends to provide more detailed information on important experimental settings. These improvements aim to enhance the clarity and completeness of our manuscript.

Comment 3: Linguistic editing would help to improve readability.

Response: Thank you for your advice. To improve the manuscript's readability, we have incorporated linguistic editing with the help of *Nature Research Editing Service*.

These revisions are dedicated to enhancing the overall clarity and fluency of the language used throughout the document.

Comment 4: Colocalization in Fig. 4a needs to be quantified. In fact the images indicate at best a partial localisation of the NWs to lysosomes.

Response: We are grateful for your invaluable suggestions. Additional analyses have been incorporated to quantify the extent of colocalization (Fig. R16a), revealing that the colocalization ratio of ICG@aAGd-NWs and lysosomes is approximately 67.1% (Fig. R16b). These findings are presented in Line 234-236 and Fig. 4a, b of the Manuscript.

Fig. R16 Colocalization analysis of cellular uptake. (a) Confocal laser scanning microscope (CLSM) images of CT26 cells stained with ICG@aAGd-NWs and Lyso-tracker Green, respectively. The white solid line represents the path used for co-localization analysis, scale bar = 2 μ m. (b) Colocalization analysis of treated CT26 tumor cells.

Comment 5: How was viability assessed in Fig. 4f and g and why does radiotherapy (RT) alone cause so little toxicity?

Response: Thank you for your inquiry. In Fig. 4f and g, cellular viability was assessed by CCK-8 assay (Dojindo, Japan). The absorbance was measured at 450 nm using a microplate reader. We have updated the figure legends to clearly specify this method for assessing cellular viability. Doses ranging from 4-10 Gy are frequently selected to induce immunogenic cell death in tumor cells and initiate anti-tumor immune responses⁷⁻¹⁰. As reported in the literature, the tumor cell inhibition rate at a radiation dose of 5 Gy ranges from 20% to 40%⁷⁻¹⁰. While Higher doses of radiotherapy are more efficacious in killing tumor cells, they also pose a risk of severe toxic side

effects.

7. Brooks, E. D., Chang, J. Y. et al. Time to abandon single-site irradiation for inducing abscopal effects. *Nat. Rev. Clin. Oncol.* **16**, 123-135 (2019).

8. Petroni, G., Cantley, L.C., Santambrogio, L. et al. Radiotherapy as a tool to elicit clinically actionable signalling pathways in cancer. *Nat. Rev. Clin. Oncol.* **19**, 114-131 (2022).

9. Lu, K. et al. Low-dose X-ray radiotherapy-radiodynamic therapy via nanoscale metal-organic frameworks enhances checkpoint blockade immunotherapy. *Nat. Biomed. Eng.* **2**, 600-610 (2018).

10. Hutchinson, L. Abscopal responses: pro-immunogenic effects of radiotherapy. *Nat. Rev. Clin. Oncol.* **12**, 504 (2015).

Comment 6: To my impression the ring-like aggregation of fluorescence all around the structure in Fig 5a seems to indicate the opposite of what the authors claim namely the inability of the NWs to penetrate the spheroids. Some clarification or positive controls would be useful.

Response: Thanks for the reviewer's constructive comments. Due to the light absorption and scattering properties inherent in most biological samples, the penetration depth of confocal laser microscopy is typically limited to approximately 100 μm^2 , with our data extending up to 70 μm . Following insightful suggestions from the reviewers, we investigated the maximum penetration capability of ultra-fine aAGd-NWs in 3D CT26 spheroids within the permissible penetration range of laser confocal microscopy (**Fig. R3a**). The experimental findings indicate that the maximum imaging depth of CLSM is confined to 120 μm . To delve deeper into the penetration capabilities (200-400 μm) of ultrafine aAGd-NWs in 3D CT26 spheroids, we dissociated these spheroids into single cells for flow cytometry analysis. As illustrated in **Fig. R3b**, compared to non-treated 3D cell spheroids (2.2%), the uptake rate of ICG@aAGd-NWs in treated tumor cells within 3D spheroids was significantly higher at 79.7%. These findings suggest that ultrafine aAGd-NWs possess the capacity to deeply penetrate the spheroids and effectively infiltrate a substantial proportion of tumor cells. These findings are presented in Line 293-302 of the

Manuscript, and are further illustrated in Supplementary Figure 5 of the Supplementary Information.

Fig. R17 Flow cytometry analysis of the digested tumor cells from spheroids treated by Vehicle or ICG@aAGd-NWs for 24 hours.

2. Jonkman, J., Brown, C. M., Wright, G. D. et al. Tutorial: guidance for quantitative confocal microscopy. *Nat. Protoc.* **15**, 1585-1611 (2020).

Comment 7: Fig. 5 d-f again indicate only very little toxicity by RT alone.

Response: Thank you for your inquiry. Doses ranging from 4-10 Gy are frequently selected to induce immunogenic cell death in tumor cells and initiate anti-tumor immune responses⁷⁻¹⁰. As reported in the literature, the tumor cell inhibition rate at a radiation dose of 5 Gy ranges from 20% to 40%⁷⁻¹⁰. While Higher doses of radiotherapy are more efficacious in killing tumor cells, they also pose a risk of severe toxic side effects.

7. Brooks, E. D., Chang, J. Y. et al. Time to abandon single-site irradiation for inducing abscopal effects. *Nat. Rev. Clin. Oncol.* **16**, 123-135 (2019).

8. Petroni, G., Cantley, L.C., Santambrogio, L. et al. Radiotherapy as a tool to elicit clinically actionable signalling pathways in cancer. *Nat. Rev. Clin. Oncol.* **19**, 114-131 (2022).

9. Lu, K. et al. Low-dose X-ray radiotherapy-radiodynamic therapy via nanoscale metal-organic frameworks enhances checkpoint blockade immunotherapy. *Nat. Biomed. Eng.* **2**, 600-610 (2018).

10. Hutchinson, L. Abscopal responses: pro-immunogenic effects of radiotherapy. *Nat. Rev. Clin. Oncol.* **12**, 504 (2015).

Comment 8: Fig. 6 h, i depict values in percent of what? It might be advisable to use another means of quantification.

Response: Thank you for bringing attention to this concern. In Fig. 6h and i, the depicted values are presented as a percentage of the vehicle as the baseline (1%), other groups are expressed as multiples of Vehicle fluorescence intensity. We acknowledge the potential ambiguity, and based on your suggestion, have adopted a times representation of fluorescence intensity in the revised manuscript (**Fig. R18**). Your feedback has been invaluable in improving the clarity of our results presentation, and we appreciate your thoughtful suggestion. These modifications are presented in Fig. 6h, i of the Manuscript.

Fig. R18 Radiosensitization efficacy of aAGd-NWs *in vivo*. (a) The relative fluorescence intensity of γ -H2A χ ($n = 3$). (b) The relative fluorescence intensity of TUNEL mean fluorescent intensity after different treatments ($n = 3$). All data were shown as mean \pm SD. Statistical significance was determined using two-tailed Student's *t*-test for pairwise comparisons, and one-way ANOVA analysis of variance for multiple groups. *p* values > 0.05 were considered non-significant (N.S.), while *p* values < 0.05 were considered statistically significant.

Comment 9: The band pattern in Fig 7 pIRF3 and pSTING is different (despite exact same molecular weight) indicating that only one loading control is not appropriate. In any case phosphonoepitope-specific antibodies should have non-phosphonoepitope-specific antibodies (used on the same stripped membrane) of

the same protein as control.

Response: We highly appreciate the reviewer's valuable suggestion. The expression levels of total IRF3 and STING are now presented in Fig. 7c of the manuscript (Fig. R19). These modifications are presented in Fig. 7d of the Manuscript.

Fig. R19 Western blot of IRF3, p-IRF, STING, p-STING, and β -actin.

Comment 10: RT has been described to emit DAMPs which is not the case in this work. This should be discussed.

Response: Thank you for highlighting this observation. We recognize that radiotherapy (RT) has been reported to release Damage-Associated Molecular Patterns (DAMPs), and we appreciate your suggestions to incorporate this discussion into our study. In the revised manuscript, this issue is specifically addressed, with provision of background and consideration of pertinent literature on this subject. In both our studies and those of others, doses of 4-10 Gy are commonly utilized to induce immunogenic death of tumor cells and initiate anti-tumor immunity^[7-10]. For instance, RT has the potential to induce CRT translocation. Furthermore, RT can lead to damage and release DNA, thus promoting the activation of the cGAS/STING pathway in adjacent immune cells and resulting in the secretion of interferon (IFN). However, the efficiency of RT in inducing the exposure or release of these DAMPs is notably limited¹¹⁻¹³, leading to seemingly inadequate capacity to initiate anti-tumor immune responses. For instance, the efficacy of RT-induced *in situ* vaccination is

deemed unsatisfactory, evidenced by abscopal effects manifesting in less than 1% of clinical cases. Consequently, there is a necessity for improved strategies to enhance RT-induced *in situ* vaccination. This information has been inserted in Line 604-613 of the Manuscript.

11. Krysko, D., Garg, A., Kaczmarek, A. et al. Immunogenic cell death and DAMPs in cancer therapy. *Nat. Rev. Cancer* **12**, 860-875 (2012).

12. Rodriguez-Ruiz, M. E., Vitale, I., Harrington, K. J. et al. Immunological impact of cell death signaling driven by radiation on the tumor microenvironment. *Nat. Immunol.* **21**, 120-134 (2020).

13. Galluzzi, L., Aryankalayil, M. J., Coleman, C. N. et al. Emerging evidence for adapting radiotherapy to immunotherapy. *Nat. Rev. Clin. Oncol.* **20**, 543-557 (2023).

Comment 11: aAGd-NWs should be benchmarked against other radiosensitizers and ICD inducers.

Response: Thank you for your suggestion. First, GGd-NCPs, comprising Gd coordinated with 5'-GMP (non-functional ligand), are designed for High-Z Radio-sensitization. By substituting the non-functional ligand 5'-GMP with ara-AMP, aAGd-NWs are synthesized to synergistically enhance High-Z radiosensitization and inhibit DNA damage repair. Furthermore, aAGd-NWs have demonstrated a superior capacity to induce ICD and initiate anti-tumor immunity.

Moreover, we recognize the importance of contrasting aAGd-NWs with other established inducers of immunogenic cell death (ICD). In the revised manuscript, classic ICD inducers such as doxorubicin and oxaliplatin are introduced. As illustrated in **Fig. R20a-d**, the capability of doxorubicin and oxaliplatin to induce ICD was comparable to RT alone, yet significantly inferior to aAGd-NW-sensitized RT. This analysis seeks to offer a thorough evaluation of aAGd-NWs' performance and bolster the overall scientific rigor of our study. We sincerely thank you again for your insightful comments. These findings are presented in Line 456-458 of the Manuscript, and are further illustrated in Supplementary Figure 9 of the Supplementary Information.

Fig. R20 ICD induction. (a) Immunofluorescence of CT26 cells stained with anti-CRT antibody, scale bar = 50 μm . (b) Quantification of relative CRT mean fluorescent intensity ($n = 3$). (c, d) Detection of cytoplasmic HMGB1 (c) and ATP secretion (d) by ELISA kit and luciferin-based ATP assay kit ($n = 3$). All data were shown as mean \pm SD. Two-tailed Student's *t*-test was used to calculate statistical differences between two groups, and one-way ANOVA analysis of variance was used for multiple groups. *p* values > 0.05 represented nonsignificance (N.S.) and *p* values < 0.05 represented statistically significant.

Special thanks to Reviewer #4 for his/her good comments. These comments have significantly improved the quality of this paper.

We tried our best to improve the manuscript and made some changes in the manuscript. These changes will not influence the content and framework of the paper. We appreciate for Reviewers' warm work earnestly, and hope that the corrections will meet with approval. Once again, thank you very much for your comments and suggestions. These comments have significantly improved the quality of this paper.

REVIEWER COMMENTS

Reviewer #1 (Remarks to the Author):

The authors have addressed most of my concerns, but skipped the most critical one: The dose of Gd in the aAGd-NWs for i.v. injection is 0.1 mmol/kg, similar to the clinical contrast agent for in vivo use. Given the potential low stability of the chelates, it should be a concern for clinical development, even the authors indicated acceptable safety in the discussions. It should be further discussed.

Also Gd based contrast agents have been used in clinical practice. The complex in this work is not a stable complex, at least not demonstrated in this work. Although the authors showed promising therapeutic outcomes with the GdNanoWire in cancer therapy. The safety of the Gd agent could be a major concern.

Reviewer #2 (Remarks to the Author):

The authors have been largely responsive to my previous comments and have performed experiments to provide additional data supporting their conclusions. I have no further questions.

Reviewer #3 (Remarks to the Author):

The authors have addressed all my concerns, the revised version could be accepted in Nat Common.

Reviewer #4 (Remarks to the Author):

The authors have discussed some concerns raised by the referees, some methodological problems remain unanswered or have even become more severe during revision.

Immunoblot protein of interest and loading controls do not seem to arise from the same blot and are therefore questionable. Full scan blots should be furnished as supplemental files. BTW in Fig. R19 molecular weights can not be right.

Calreticulin immunofluorescence clearly depicts intracellular but not membrane exposed protein.

Single cell dissociation as a means of showing compound penetration into 3d structures has an extreme risk of carry over. Histological embedding and sample preparation for and re-imaging would be much better.

For colocalization studies one needs to exclude any kind of filter bleed through. Meaning lysosomal staining and ICG@aAGd-NW treatment need to be conducted separately and imaged. Btw this should be LysoTracker not Lysotraker. A real colocalization study should be conducted yielding a colocalization coefficient instead of measuring fluorescence along a single line. There is free software available that can do that.

Response to Referees

Dear Reviewers:

Thank you for your constructive comments concerning our manuscript entitled “Chiral Coordination Polymer Nanowires Boost Radiation-Induced *In Situ* Tumor Vaccination” (ID: NCOMMS-23-28217A). These comments are all invaluable and have been instrumental in the revision and enhancement of our paper, providing significant guidance for our research. We have examined the comments carefully and made corrections that we hope will meet with your approval. Revisions are highlighted in yellow in the paper. The point-to-point responses to the reviewer's comments are as following:

Reviewer #1 (Gd-based nanomaterials):

Comment 1: The authors have addressed most of my concerns, but skipped the most critical one: The dose of Gd in the aAGd-NWs for i.v. injection is 0.1 mmol/kg, similar to the clinical contrast agent for *in vivo* use. Given the potential low stability of the chelates, it should be a concern for clinical development, even the authors indicated acceptable safety in the discussions. It should be further discussed.

Also Gd based contrast agents have been used in clinical practice. The complex in this work is not a stable complex, at least not demonstrated in this work. Although the authors showed promising therapeutic outcomes with the Gd NanoWire in cancer therapy. The safety of the Gd agent could be a major concern.

Response: We appreciate your thorough review and constructive comments on our manuscript. We understand your concern about the dose of Gd in aAGd-NWs being similar to clinical contrast agents for *in vivo* use. We agree that it is essential to thoroughly discuss and justify the dosage, especially considering the potential low stability of the chelates.

Therefore, we propose to discuss the safety of aAGd-NWs from the following aspects: (1) The toxicity of Gd is primarily attributed to free Gd, as evidenced by the results from Supplementary Figure 19a. However, it should be noted that the ultimate release of free Gd requires a pH lower than 3.8, which is a condition almost unattainable in

human metabolic processes. Preliminary studies conducted by our team demonstrated that gadolinium and 5'-GMP ligand form nanoscale coordination polymers, which are predominantly excreted through urine in a coordinated form rather than as free agents¹⁴. Consequently, aAGd-NWs exhibited no significant nephrotoxicity, even when the total dosage exceeded that of free gadolinium by more than fivefold.

(2) Unlike the use of contrast agents in the general population, aAGd-NWs are specifically targeted for cancer treatment, often allowing for a broader tolerance in terms of safety. Typically, the clinical dose of gadolinium-based contrast agents, such as Magnevist, is approximately 0.1 mmol kg⁻¹, which has been shown to exhibit reliable safety profiles. Extrapolating based on body surface area⁵¹, the dosing range of Gd for mice is about 1.0~1.2 mmol kg⁻¹, comparable to the dosage of gadolinium-based contrast agents used in mice for MRI purposes^{52,53}. The cumulative dose of Gd (0.1 mmol kg⁻¹ × 2) we employed is obviously lower than the single dose for Gd-based contrast agents. Findings from acute toxicity studies further demonstrate that even when the total dosage increased from 0.1 mmol kg⁻¹ × 2 to 0.1 mmol kg⁻¹ × 4, aAGd-NWs continued to exhibit excellent safety profiles *in vivo* (Supplementary Figure 19b, c). We have added these discussions into the Manuscript (Page 36-38, Line 676-715).

References of Manuscript

14. Huang, Z., Wang, Y., Yao, D. et al. Nanoscale coordination polymers induce immunogenic cell death by amplifying radiation therapy mediated oxidative stress. *Nat. Commun.* **12**, 145 (2021).
51. Davidson, I. W. F., Parker, J. C. & Beliles, R. P. Biological basis for extrapolation across mammalian species. *Regul. Toxicol. Pharmacol.* **6**, 211-237 (1986).
52. Tweedle, M. F., Wedeking, P., Kumar, K. Biodistribution of radiolabeled, formulated gadopentetate, gadoteridol, gadoterate, and gadodiamide in mice and rats. *Invest. Radiol.* **30**, 372-380 (1995).
53. Kromrey, M. L., Oswald, S., Becher, D. et al. Intracerebral gadolinium deposition following blood – brain barrier disturbance in two different mouse models. *Sci. Rep.* **13**, 10164 (2023).

Special thanks to Reviewer #1 for his/her good comments. These comments have undoubtedly strengthened the overall quality and clarity of our manuscript.

Reviewer #4 (Cancer Immunology, ICD):

Comment 1: The authors have discussed some concerns raised by the referees, some methodological problems remain unanswered or have even become more severe during revision.

Response: We appreciate your continued engagement with our manuscript and your insightful feedback on the revisions. We recognize and take seriously your comments regarding the methodological concerns and pledge to diligently address these issues to further refine the overall quality of our work.

Comment 2: Immunoblot protein of interest and loading controls do not seem to arise from the same blot and are therefore questionable. Full scan blots should be furnished as supplemental files. BTW in Fig. R19 molecular weights can not be right.

Response: We are grateful for your insightful comments and suggestions regarding the immunoblot data. We appreciate your attention to detail. To address this issue, we have included full-scan blots in Source Data, and corrected the predicted molecular weight information for Fig. 7d (IRF-3 48 Kda, *p*-IRF-3 48 Kda; STING 37 Kda, *p*-STING 40 Kda, β -actin 42 Kda) (Page 24, Line 436-437).

Fig. 7 Enhanced DNA damage and ICD induction. (d) Western blot of IRF3, *p*-IRF3, STING, *p*-STING, and β -actin as an internal reference.

Source Data of Fig. 7d. Full-scan western blots of IRF3, *p*-IRF3, STING, *p*-STING, and β -actin as an internal reference (Source Data File).

Comment 3: Calreticulin immunofluorescence clearly depicts intracellular but not membrane exposed protein.

Response: We appreciate your keen observation and thank you for bringing attention to the concern regarding the calreticulin immunofluorescence in our manuscript. Upon reviewing the results, we concur with the reviewers that the originally presented images did not exhibit typical characteristics. Therefore, we have repeated the experiments and obtained new images. As shown in Fig.7f, the membrane-exposed CRT exhibits an extranuclear distribution, which differs from the ring-like structure of the membrane displayed by dyes such as DIR. This observation aligns with findings from multiple published studies^{1,2}. The updated Fig. 7f has now been incorporated into the manuscript (Page 24, Line 438-440).

Fig. 7 Enhanced DNA damage and ICD induction. (f) Immunofluorescence of CT26 cells stained with anti-CRT antibody, scale bar = 20 μ m.

References

1. Zhang, Y., Yang, S., Yang, Y. *et al.* Resveratrol induces immunogenic cell death of human and murine ovarian carcinoma cells. *Infect. Agents. Cancer* **14**, 27 (2019).
2. Menger, L., Vacchelli, E., Adjemian, S. *et al.* Cardiac glycosides exert anticancer effects by inducing immunogenic cell death. *Sci. Transl. Med.* **4**,143ra99 (2012).

Comment 4: Single cell dissociation as a means of showing compound penetration into 3d structures has an extreme risk of carry over. Histological embeddin and sample preparation for and re-imaging would be much better.

Response: We are grateful for your thoughtful consideration of our experimental approach for assessing aAGd-NWs penetration into 3D structures. Due to an oversight, we omitted details of the method for the single-cell dissociation experiment. Therefore, we would like to elaborate on our experimental approach: The dissociated single-cell suspension was subjected to three washes with PBS and subsequently collected by short-duration and low-speed centrifugation (2000 rpm × 5 min) to effectively remove unuptaked ICG@aAGd-NWs. Ultimately, the cells treated with ICG@aAGd-NWs were resuspended in PBS for flow cytometry analysis. Furthermore, uniformly dispersed ICG@aAGd-NWs were collectable exclusively under long-duration and high-speed centrifugation conditions (12000 rpm × 30 min) and were not retrievable under short-duration and low-speed centrifugation conditions (2000 rpm × 5 min). Consequently, ICG@aAGd-NWs that are not internalized will be effectively removed, ensuring their non-persistence in the samples for detection. We appreciate the reviewer's carefulness, and supplement the method in the Manuscript (Page 46-48, Line 895-901).

Comment 5: For colocalization studies one needs to exlcude any kind of filter bleed through. Meaning lysomal staining and ICG@aAGd-NW treatment need to be conducted seperatly and imaged. Btw this should be LysoTracker not Lysotraker. A real colocalization study should be conducted yielding a colocalization coeffizient instead of measunging fluorescence along a single line. There is free software available that can do that.

Response: We are grateful for your insightful comments regarding colocalization studies in our manuscript. We have included detailed information on the revised experimental setup to ensure the accuracy of our colocalization analysis. The accurate term "LysoTracker" has been accurately incorporated throughout the manuscript. To further enhance the rigor of our colocalization studies, images are now analyzed using

ImageJ (Colocalization Finder) for the calculation of the Pearson Correlation Coefficient (PCC = 0.8673). In the revised manuscript, a comprehensive description of the colocalization analysis methodology is provided (Page 44, Line 841-850), including colocalization coefficients to quantitatively assess the overlap between lysosomes and ICG@aAGd-NWs (Page 14, Line 251-256).

Fig. 4 Radiosensitization of aAGd-NWs *in vitro*. (a) Confocal laser scanning microscope (CLSM) images of CT26 cells stained with DAPI, LysoTracker Green and ICG@aAGd-NWs, respectively. The co-localization of aAGd-NWs within lysosomes is indicated by the yellow regions, scale bar = 20 μm . (b) Pearson correlation colocalization (PCC) analysis of treated CT26 tumor cells by *ImageJ* (Colocalization Finder).

Special thanks to Reviewer #4 for his/her good comments. These comments have significantly improved the quality of this paper.

We tried our best to improve the manuscript and made some changes in the manuscript. These changes will not influence the content and framework of the paper. We appreciate for Reviewers' warm work earnestly, and hope that the corrections will meet with approval. Once again, thank you very much for your comments and suggestions. These comments have significantly improved the quality of this paper.

REVIEWERS' COMMENTS

Reviewer #1 (Remarks to the Author):

The authors have properly addressed the concerns.

Reviewer #4 (Remarks to the Author):

With the present revision the authors have further improved their manuscript. They have furnished convincing evidence pleading in favor of their hypothesis and have answered to all questions posed by the reviewers.